EMBO
Molecular Medicine

# Restricting SLC7A5-mediated Leucine uptake in T cells prevents acute GVHD and maintains GVT response

Nieves Fernández-Gallego [ID] [1,2,8], Blanca Anega [ID] [1,2,8], Susana Luengo-Arias [ID] [1,2], Maider Bizkarguenaga [3], Rubén Gil-Redondo [ID] [3], Nieves Embade [ID] [3], Laura Navarrete-Arias [1], Marta Ramírez-Huesca [2], Emigdio Álvarez-Corrales [ID] [4], Sara G Dosil [ID] [1,2], Raquel Castillo-González [ID] [1,2,4], Amelia Rojas-Gomez [ID] [1,2], Inés Espeleta [1], Sara Martínez-Martínez [5,6], Arantzazu Alfranca [ID] [1,5], Virginia G de Yebenes [ID] [4], Noa Beatriz Martín-Cófreces [ID] [1,5], Julián Aragonés [ID] [5,7], Pilar Martin [ID] [2,5], Oscar Millet [ID] [3], Francisco Sánchez-Madrid [ID] [1,2,5] & Danay Cibrian [ID] [1,2,5✉]

## Abstract

The L-Leu amino acid transporter SLC7A5 has become an important target in inflammation and cancer. However, its role in acute graft-versus-host disease (aGVHD) and graft versus tumor (GVT) remains unexplored. We demonstrate that SLC7A5 deletion affected T cell activation, expansion and survival, and reduced IFNγ and granzyme B expression, thus controlling aGVHD, but without effect on tumor growth. On the other hand, dietary restriction of L-Leu reduced aGVHD by controlling T cell expansion, inducing apoptosis, and affecting granzyme B secretion. However, CD8 T cells did not fail to activate and express IFNγ in the absence of L-Leu, and showed an increased proportion of central memory T cells, which contributed to the GVT response. Deletion of SLC7A5 in T cells compromises mTORC1, glycolysis and mitochondrial oxidation. On the contrary, L-Leu removal reduced mTORC1 and completely blocked glycolysis but preserved mitochondrial function, favoring the generation of central memory responses and expression of stemness marker TCF1. In addition, our metabolomics data underscores the L-Leu-derived metabolite β-hydroxybutyrate as an important marker for SLC7A5-dependent allogenic T cell expansion in aGVHD.

**Keywords** Graft-versus-Host Disease; Graft-versus-Tumor; SLC7A5; L-Leu; T-Cell-Metabolism
**Subject Categories** Immunology; Metabolism

## Introduction

Allogenic hematopoietic cell transplantation (allo-HCT) is a therapeutic approach for the treatment of malignant and non-malignant diseases, but the risk of graft-versus-host disease (GVHD) limits the widespread use of this procedure (DeFilipp et al, 2021; Ferrara et al, 2009). Acute GVHD (aGVHD) is initiated by alloreactive donor αβ TCR[+] T cells activated against recipient-specific antigens (Zeiser and Blazar, 2017). Pre-transplant conditioning treatments, including irradiation, induce severe damage to intestinal epithelial cells. As a result of gut barrier disruption, there is an increase in pathogen translocation that enhances the inflammatory response mediated by the production of danger signals, including danger-associated molecular patterns (DAMPs) and proinflammatory cytokines such as TNF-α, IL-1, and IL-6, and the increased expression of adhesion molecules, MHC antigens, and costimulatory molecules (Chen et al, 2009; Jankovic et al, 2013; Taylor et al, 2005). Therefore, the prevention of intestinal epithelial injury is particularly relevant to prevent aGVHD (Jansen et al, 2022; Mathewson et al, 2016).

T cell activation and proliferation involve increased protein synthesis, which relies on enhanced amino acid (AA) uptake (Chapman et al, 2020). The expression of L-type amino acid transporter 1 (LAT1, SLC7A5) is upregulated in mouse and human-activated T cells, where it constitutes the major transporter for large and neutral AA (L-Leu, L-Ile, L-Val) and aromatic AA (L-Trp, L-Phe) (Fotiadis et al, 2013; Hayashi et al, 2013; Sinclair et al, 2013). SLC7A5 forms a heterodimer complex with SLC3A2 (CD98) through a disulfide bond, which is required for its transport and stability in the plasma membrane (Cormerais et al, 2016). SLC7A5

[1]Department of Immunology, Instituto de Investigación Sanitaria Hospital Universitario de La Princesa (IIS-Princesa), Universidad Autónoma de Madrid (UAM), Madrid, Spain. [2]Centro Nacional de Investigaciones Cardiovasculares (CNIC), Madrid, Spain. [3]Precision Medicine and Metabolism Laboratory, CIC bioGUNE, Parque Tecnológico de Bizkaia, Ed. 800, 48160 Derio, Spain. [4]Department of Immunology, Ophthalmology and Ear, Nose and Throat (ENT), Complutense University, School of Medicine and Instituto de Investigación Sanitaria Hospital 12 de Octubre (imas12), Madrid, Spain. [5]Centro de Investigación Biomédica en Red de Enfermedades Cardiovasculares (CIBERCV), Instituto de Salud Carlos III, Madrid, Spain. [6]Gene Regulation in Cardiovascular Remodeling and Inflammation Group, Centro Nacional de Investigaciones Cardiovasculares (CNIC), Madrid, Spain. [7]Research Unit, Hospital Santa Cristina, Instituto de Investigación Sanitaria, Universidad Autónoma de Madrid (UAM), Madrid, Spain. [8]These authors contributed equally: Nieves Fernández-Gallego, Blanca Anega. ✉E-mail: dcibrianlab@gmail.com; danay.cibrian.externo@salud.madrid.org; danay.cibrian@externo.cnic.es

is expressed in highly proliferative cells, including tumoral cells, where it controls mTOR activation and proliferation by mediating L-Leu uptake (Cibrian et al, 2020; Elorza et al, 2012; Hayashi et al, 2020; Sinclair et al, 2013). Notably, SLC7A5 is overexpressed in human T-cell acute lymphoblastic leukemia/lymphoma cells and non-Hodgkin lymphoma, which are frequent indications for HCT (Jigjidkhorloo et al, 2021; Passweg et al, 2023; Rosilio et al, 2015). Accordingly, SLC7A5 has emerged as a promising novel strategy for cancer therapy and to control inflammatory and immunological disorders such as psoriasis, rheumatoid arthritis, or allergic diseases (Cibrian et al, 2020; Hayashi et al, 2020; Oda et al, 2010; Ogbechi et al, 2023; Owada et al, 2022).

Global immunosuppression is the mainstay of therapy for aGVHD (Martin et al, 2012). However, its effectiveness is partial, and the complications of chronic immunosuppression are detrimental to the patients (Carpenter et al, 2015). In addition, immunosuppression regimens entail the loss of the graft-versus-tumor (GVT) effect, increasing the risk of tumor escape and relapse (Carpenter et al, 2015; Martin et al, 2012). The development of novel approaches that prevent GVHD while preserving GVT activity remains a long-sought goal. The relevance of immunometabolism in modulating alloreactive donor T cell responses to control GVHD and promote GVT has been highlighted by several HCT studies (Mohamed et al, 2021). Aside from pharmacological targeting of cellular metabolism, there is a growing appreciation of the potential of direct modulation of environmental cues in fine-tuning cell metabolism during inflammation and cancer (Limpert et al, 2023; McIntyre et al, 2023). Diet deprivation or supplementation of specific nutrients or metabolites can exert important regulatory effects on immune responses and cancer expansion, thus providing novel opportunities for nutritional therapeutic intervention in aGVHD (Crowther et al, 2009; Noth et al, 2013; Pereira et al, 2020; Sheen et al, 2011).

Here we demonstrate that targeted deletion of SLC7A5 in αβ T cells in a mouse model of full mismatch effectively protects against aGVHD but does not maintain GVT response. Furthermore, our data provides evidence that reducing L-Leu intake through dietary modification could be a useful strategy to control immune-mediated injury after allogenic HCT, to prevent inflammation-induced metabolic alterations, and to avoid malignant cell expansion.

# Results

## Genetic deletion of SLC7A5 in T cells efficiently prevents aGVHD-associated colon inflammation and mortality

We first determined whether SLC7A5 deletion in T cells controls the mortality in a full MHC-mismatched model of aGVHD. For this purpose, transgenic mice expressing CD45.1 haplotype or β-actin-eGFP as bone marrow (BM) donors and SLC7A5$^{WT}$ or SLC7A5$^{\Delta CD4}$ mice expressing Tomato (Tmt) as splenic T cells (STC) donors were used (Fig. EV1A). All BALB/c mice receiving STC from SLC7A5$^{WT}$ (SLC7A5$^{WT}$ → BALB/c) died with evident clinical signs of aGVHD, including progressive weight loss, severe diarrhea, hunched back, ruffled fur and, in some mice, hair loss and macroscopic skin damage. In contrast, 100% of mice receiving STC from SLC7A5$^{\Delta CD4}$

(SLC7A5$^{\Delta CD4}$ → BALB/c) survive for at least 80 days, with no macroscopic signs of aGVHD (Figs. 1A and EV1B). Significant body weight reduction was detected as early as 7 days post-transplant in SLC7A5$^{WT}$ → BALB/c mice, while the weight of SLC7A5$^{\Delta CD4}$ → BALB/c mice was not altered (Fig. 1B). Autopsies also showed evidence of colon shortening after 7 days of transplantation in BALB/c mice receiving SLC7A5$^{WT}$ T cells, whereas colon length in SLC7A5$^{\Delta CD4}$ → BALB/c was not affected (Fig. 1C). Colon inflammation hallmarks, including transmural leukocyte infiltration, loss of goblet cells, and focal areas of mucosal erosion, were already observed in SLC7A5$^{WT}$ → BALB/c mice as early as 7 days post-transplant. Signs of colon alterations were absent in SLC7A5$^{\Delta CD4}$ → BALB/c mice at day 7 post-transplant (Fig. 1D). Histopathological observations confirmed exacerbated injury and inflammation in SLC7A5$^{WT}$ → BALB/c colon sections at 40 days post-transplant, including severe erosion of crypts, complete loss of goblet cells, and high mononuclear cell infiltrate. Remarkably, colon sections from SLC7A5$^{\Delta CD4}$ → BALB/c mice showed unaltered architecture without signs of inflammation after 40 days of transplant (Fig. 1E). Increased neutrophil (CD11b$^+$Ly6C$^+$Ly6G$^+$) and monocyte (CD11b$^+$Ly6C$^+$Ly6G$^-$) infiltration was also detected by flow cytometry in the colon of SLC7A5$^{WT}$ → BALB/c at 7 days post-transplant, as compared to a very low infiltration in SLC7A5$^{\Delta CD4}$ → BALB/c (Figs. 1F and EV1C).

## SLC7A5 is required for alloreactive T cell infiltration in lymphoid organs and aGVHD-target tissues

Next, we analyzed the absolute numbers and distribution of CD4$^+$ and CD8$^+$ T cell populations in recipient mice 7 days after transplantation. The origin of αβ T cells after transplant was identified by the expression of GFP or Tmt in gating strategies (Fig. EV1D). The total number of allogenic CD4$^+$Tmt$^+$ and CD8$^+$Tmt$^+$ cells was significantly higher in SLC7A5$^{WT}$ → BALB/c than in SLC7A5$^{\Delta CD4}$ → BALB/c in mesenteric lymph nodes (mLN) and spleen (Fig. 1G,H). Attending to T cell origin, most CD4$^+$ and CD8$^+$ T cells correspond to Tmt$^+$ cells in lymphoid organs of SLC7A5$^{WT}$ → BALB/c mice, whereas GFP$^+$ T cells and host T cells were higher in SLC7A5$^{\Delta CD4}$ → BALB/c mice (Figs. 1G,H and EV1E). Hence, the expansion of alloreactive SLC7A5$^{WT}$ STC hinders the engraftment of GFP$^+$ BM cells, whereas the control of inflammation by genetic deletion of SLC7A5 in alloreactive T cells improves the engraftment of non-alloreactive GFP$^+$ BM cells. Immunofluorescence images of spleen sections from SLC7A5$^{WT}$ → BALB/c mice confirmed that most CD3$^+$ cells were Tmt$^+$, suggesting an expansion of this population that is not observed in SLC7A5$^{\Delta CD4}$ → BALB/c spleen sections (Fig. 1I).

Likewise, the analysis of the colon showed a higher number of CD4$^+$Tmt$^+$ and CD8$^+$Tmt$^+$ cells and a greater proportion of STC-derived T cells in SLC7A5$^{WT}$ → BALB/c than in SLC7A5$^{\Delta CD4}$ → BALB/c mice (Fig. 2A). Immunofluorescence images confirmed that T cell infiltration is much lower in colon sections from SLC7A5$^{\Delta CD4}$ → BALB/c, while colon sections from SLC7A5$^{WT}$ → BALB/c mice were prominently infiltrated with Tmt$^+$CD3$^+$ T cells (Fig. 2B). Similar results were observed in other tissues such as skin and lung (Figs. 2C,D and EV2A). In addition, we performed a competitive migration assay (Fig. EV2B) in which equal numbers of SLC7A5$^{WT}$ (CD45.1$^+$) and SLC7A5$^{\Delta CD4}$ (Tmt$^+$) cells were injected into BALB/c recipients after irradiation. The

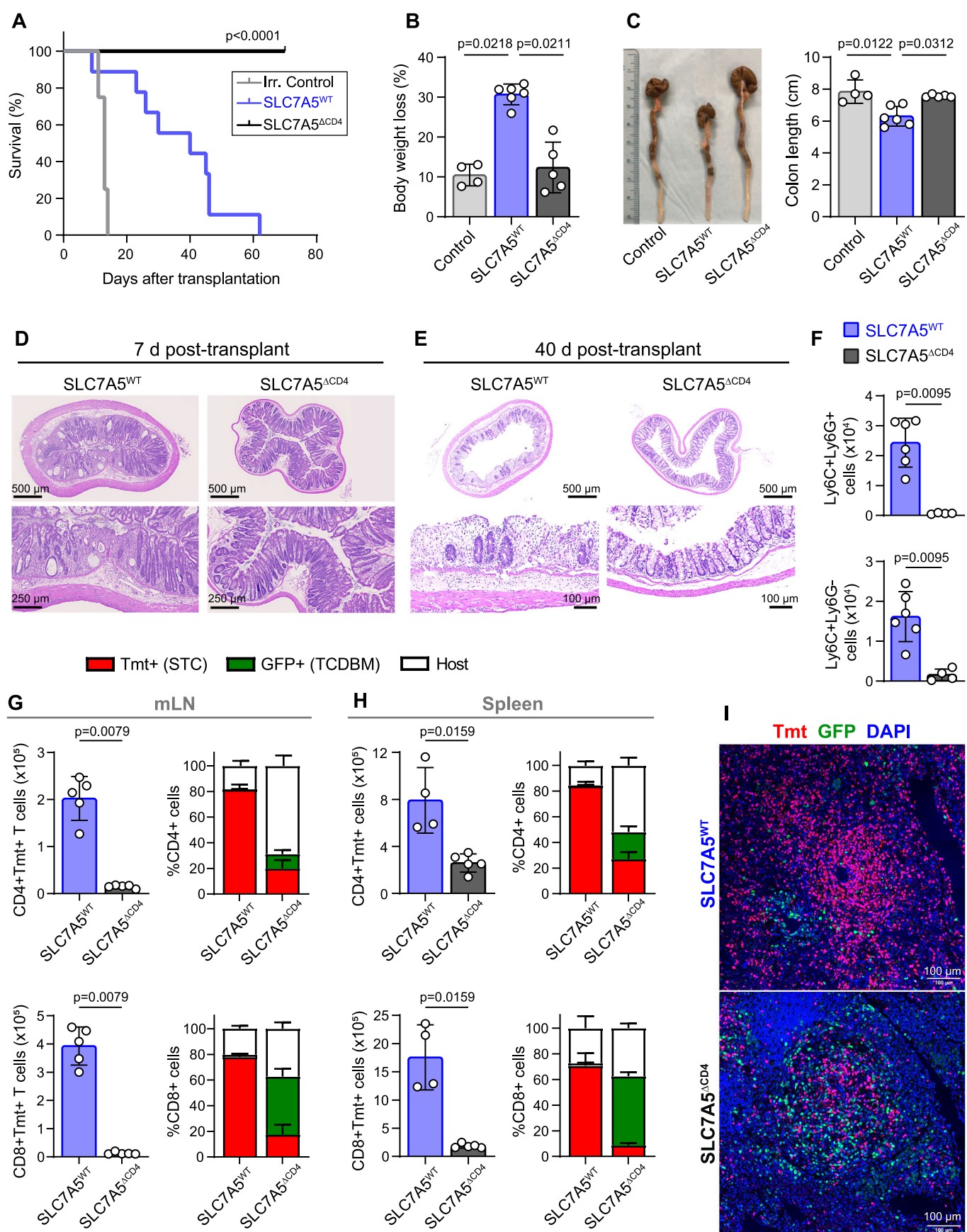

**Figure 1. Genetic deletion of SLC7A5 in donor αβ T cells prevents aGVHD.**

(A) Survival curve of irradiated control BALB/c mice (irradiated and non-transplanted controls, gray, $N = 4$) and BALB/c recipient mice transplanted with TCDBM from CD45.1+ donors and STC from SLC7A5$^{WT}$ (blue, $N = 9$) or SLC7A5$^{\Delta CD4}$ (black, $N = 10$) donors. A pool of two independent experiments is shown. Long-rank (Mantel-Cox) and Cehan-Breslow-Wilcoxon test. (B) Percentage of body weight loss at 7 d post-transplant. A representative experiment of three is shown. (C) Representative images of colon length (left) and its quantification (right) of irradiation controls, SLC7A5$^{WT} \to$ BALB/c and SLC7A5$^{\Delta CD4} \to$ BALB/c mice at 7 d post-transplant. A representative experiment of two is shown. (D, E) Representative H&E-sections of the colon from SLC7A5$^{WT} \to$ BALB/c and SLC7A5$^{\Delta CD4} \to$ BALB/c mice at 7 d (D) and 40 d (E) post-transplant. Scale bars = 500 μm in wide-field images and 250 or 100 μm in zoom areas. (F) Total number of neutrophils (Ly6C+Ly6G+) and monocytes (Ly6C+Ly6G-) in colon of SLC7A5$^{WT} \to$ BALB/c and SLC7A5$^{\Delta CD4} \to$ BALB/c mice at 7 d post-transplant. (G, H) Absolute numbers (left) of Tmt+CD4+ (top) and Tmt+CD8+ (bottom) cells and frequency (right) of Tmt+ (STC origin), GFP+ (BM origin) and double negative (host origin) T cells of total CD4+ (top) and CD8+ (bottom) cells from mesenteric lymph nodes (mLN) (G) and spleen (H) from SLC7A5$^{WT} \to$ BALB/c and SLC7A5$^{\Delta CD4} \to$ BALB/c mice at 7 d post-transplant. A representative experiment out of three is shown. (I) Representative immunofluorescence of Tmt (red) and GFP (green) in spleen sections. Nuclei were stained with DAPI (blue). Scale bars = 100 μm. (B, C, F, G, H): $N = 4$–6 per group, mean ± SD, Kruskal–Wallis test with Dunn's post-test (B, C) or two-tailed Mann–Whitney's test (F, G, H). STC splenic T cell, TCDBM T-cell-depleted bone marrow, Tmt Tomato. Source data are available online for this figure.

results confirmed the reduced migration of SLC7A5$^{\Delta CD4}$ cells in comparison to SLC7A5$^{WT}$ cells in lymphoid organs (Fig. EV2C,D) and non-lymphoid tissues (Figs. 2E and EV2D) 7 days after transplant. Altogether, these results demonstrate that SLC7A5 deletion in αβ T cells reduced the infiltration of alloreactive T cells in lymphoid and mucosal tissues. As a result, the inflammation, tissue damage and mortality associated with aGVHD are prevented.

## Dietary restriction of L-Leu controls mortality in the aGVHD mouse model

SLC7A5 is the main carrier of the essential AA L-Leu in activated T cells, which mediates the activation of the mTORC1 pathway (Napolitano et al, 2015; Nicklin et al, 2009). Hence, we considered whether the removal of L-Leu from the environment could be an alternative to the genetic deletion of SLC7A5 in allogenic T cells (Fig. EV3A). Mice fed with an L-Leu-deprived diet showed a significantly reduced incidence of mortality and a better clinical condition without signs of aGVHD, while regular-diet-fed mice showed weight loss, severe diarrhea, ruffled fur, and hunched back (Figs. 3A and EV3B). Importantly, the total number of Tmt+CD4+ and Tmt+CD8+ SLC7A5$^{WT}$ T cells was significantly reduced in the spleen of mice fed with L-Leu-deprived diet as compared to the regular diet group 7 d after transplant (Fig. 3B).

The expression of CD44 and CD62L was analyzed 7 days after transplant to explore the activation and memory response of SLC7A5$^{WT}$ allogenic T cells in (-)Leu diet-fed mice, and of SLC7A5$^{\Delta CD4}$ and SLC7A5$^{WT}$ T cells simultaneously injected in the same mice fed with regular diet (Figs. 3C and EV3C). Interestingly, SLC7A5 genetic deletion impaired in vivo activation of CD4 and CD8 T cells, observed as an increased fraction of naive T cells (CD44-CD62L+) and a reduced proportion of the effector memory T cell fraction (CD44+CD62L-), compared to SLC7A5$^{WT}$ cells (Fig. 3C). Importantly, dietary restriction of L-Leu did not affect activation of allogenic SLC7A5$^{WT}$ CD4 and CD8 T cells, as we detected more than 90% of allogenic T cells as CD44+CD62L-, similar to the activation of SLC7A5$^{WT}$ cells in regular-diet fed mice. Interestingly, a higher percentage of CD69 was detected in Tmt+CD4+ and Tmt+CD8+ of SLC7A5$^{\Delta CD4}$ T cells, as compared to SLC7A5$^{WT}$ T cells in mice fed with regular diet (Figs. 3D and EV3D). In addition, the absence of Leu in the diet also increased the percentage of CD69+ allogenic T cells, mainly in CD4 T cells, suggesting that the increase of CD69 is a compensatory response induced by nutrient deprivation.

## SLC7A5 deletion and L-Leu restriction reduce proliferation and promote apoptosis of allogenic CD4 T cells in aGVHD

The reduced absolute number of Tmt+ T cell populations in SLC7A5$^{\Delta CD4} \to$ BALB/c mice could be caused by an impaired proliferation of SLC7A5$^{\Delta CD4}$ donor cells and/or increased cell death rate. Therefore, cell proliferation was assessed at 3 days post-transplant by staining with CellTrace Violet (CV). The percentage of CV loss was higher in Tmt+CD4+ and Tmt+CD8+ cells of SLC7A5$^{WT} \to$ BALB/c compared with SLC7A5$^{\Delta CD4} \to$ BALB/c (Fig. 3E), indicating that proliferative capacity of SLC7A5$^{\Delta CD4}$ T cells is reduced. Likewise, we performed in vivo measurement of proliferation with CV after 3 days of transplant in mice fed with regular or (-)Leu diet (Fig. 3F). Our data showed that dietary restriction of L-Leu was sufficient to prevent the allogenic Tmt+CD4+ and Tmt+CD8+ T cell expansion, as achieved by genetic deletion of SLC7A5, and thus dampening aGVHD.

To assess cell death in vivo, we performed the analysis of annexin V/7-amino-actinomycin (AnnV/7-AAD) staining at 7 days post-transplant (Figs. 3G and EV3E). Tmt+SLC7A5$^{\Delta CD4}$ T cells and Tmt+SLC7A5$^{WT}$ cells injected in (-)Leu diet-fed mice showed reduced cell viability, detected by increased AnnV percentages, compared to Tmt+SLC7A5$^{WT}$ T cells transplanted to regular diet-fed mice (Fig. 3G). Importantly, CD4 T cells were more sensitive to induction of apoptosis by blocking SLC7A5-mediated L-Leu uptake (Fig. 3G). On the contrary, apoptosis of CD8 T cells was reduced or not affected by genetic deletion of SLC7A5 or L-Leu withdrawal, respectively (Fig. 3G).

These results indicate that restriction of L-Leu-uptake through SLC7A5 in allogenic T cells is sufficient to control T lymphocyte expansion induced by alloantigen reaction, which underscores the immunosuppressive potential of the AA transporter SLC7A5 and the restriction of Leu in aGVHD.

## L-Leu deprivation and SLC7A5 deletion regulate cytokine secretion, protein synthesis and mTORC1 pathway

IFNγ plays complex yet essential roles in aGVHD and GVT response (Lu and Waller, 2009). The expression of Tbet and mainly IFNγ was diminished in Tmt+ CD4 and CD8 T cells of SLC7A5$^{\Delta CD4} \to$ BALB/c compared to Tmt+ T cells of SLC7A5$^{WT} \to$ BALB/c at 7 days after transplant (Fig. 4A,B). We also explored the secretion of IFNγ and granzyme B (GzmB) in

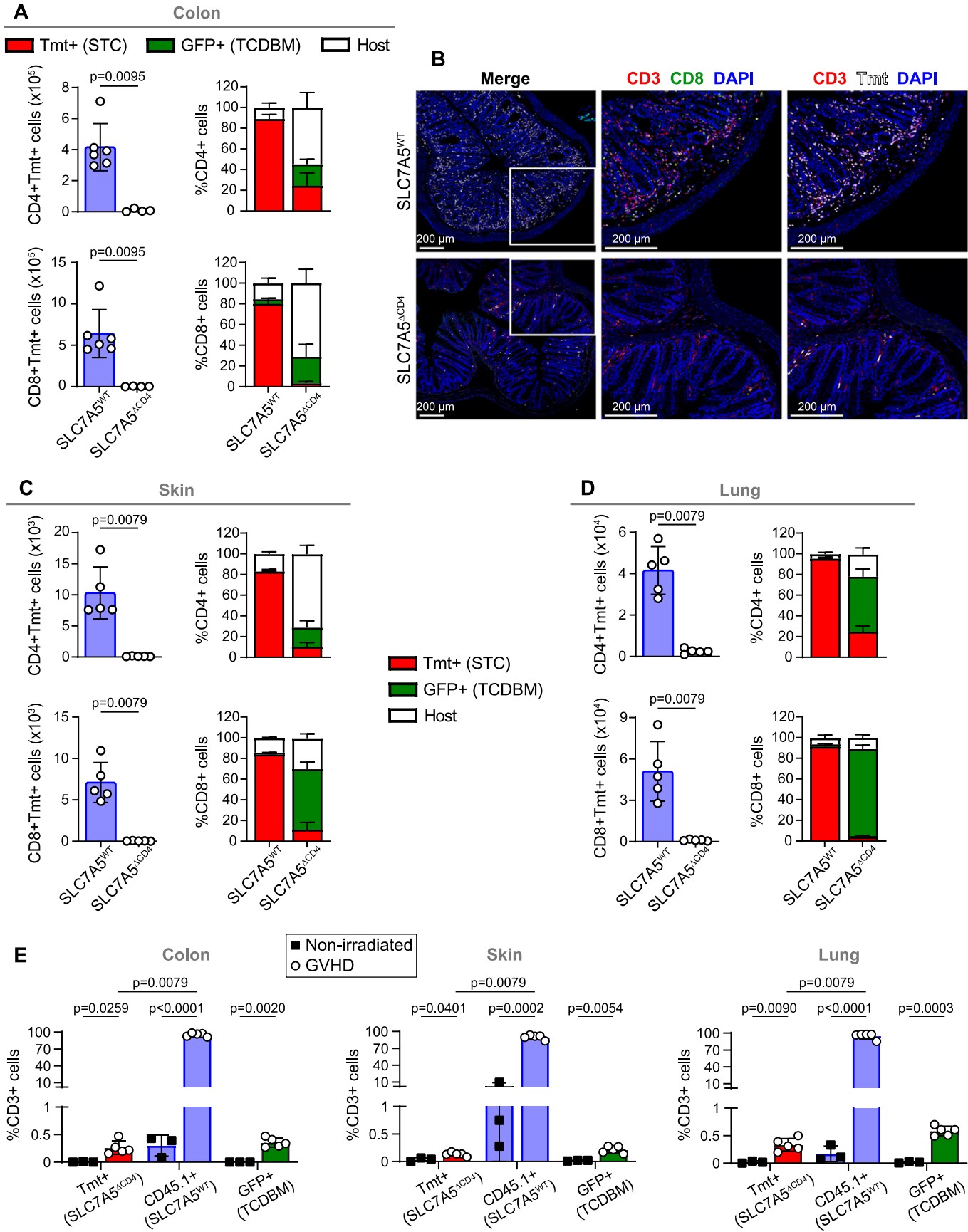

◄ **Figure 2. SLC7A5 deficiency controls alloreactive T cell infiltration in aGVHD-target tissues.**

(A) Absolute numbers (left) of Tmt+CD4+ (top) and Tmt+CD8+ (bottom) cells and frequency (right) of Tmt+ (STC origin), GFP+ (BM origin) and double negative (host origin) T cells of total CD4+ (top) and CD8+ (bottom) T cells from colon of SLC7A5WT → BALB/c and SLC7A5ΔCD4 → BALB/c mice at 7 d post-transplant. (B) Representative immunofluorescence of CD3 (red), CD8 (green), and Tmt (white) in colon sections. Nuclei were stained with DAPI (blue). All scale bars = 200 μm. (C, D) Absolute numbers (left) of Tmt+CD4+ (top) and Tmt+CD8+ (bottom) cells and frequency (right) of Tmt+ (STC origin), GFP+ (BM origin) and double negative (host origin) T cells of total CD4+ (top) and CD8+ (bottom) T cells from skin (C), and lung (D) of SLC7A5WT → BALB/c and SLC7A5ΔCD4 → BALB/c mice at 7 d post-transplant. (E) Percentages of CD3+ T cells expressing Tmt+ (SLC7A5ΔCD4 STC, red), CD45.1+ (SLC7A5WT B6.SJL STC, blue), or GFP+ (β-actin-eGFP TCDBM, green) in colon, skin, and lung from non-irradiated (control) and irradiated (GVHD) transplanted BALB/c recipient mice in a competitive migration assay. N = 3–5 per group, mean ± SD, Kruskal–Wallis test with Dunn's post-test for control and GVHD comparison or two-tailed Mann–Whitney's test for Tmt+ and CD45.1+ of GVHD group comparison. (A, C, D): N = 4–6 per group, mean ± SD, two-tailed Mann–Whitney's test. STC splenic T cell, TCDBM T-cell-depleted bone marrow, Tmt Tomato. Source data are available online for this figure.

in vitro concanavalin A-activated CD8+ T cells from SLC7A5ΔCD4 mice and SLC7A5WT mice, using complete or (-)Leu medium (Fig. 4C). In this setting, CD8 T cells from SLC7A5ΔCD4 mice showed a clear reduction in the secretion of IFNγ (Fig. 4C). In contrast, the expression of IFNγ by CD8 T cells was not affected by Leu removal (Fig. 4C). In addition to IFNγ, the expression of GzmB by CD8 T cells is essential to cause lethal GVHD (Graubert et al, 1996). Both genetic deletion of SLC7A5 or activation in (-)Leu medium significantly affect the expression of GzmB in CD8 T cells (Fig. 4C), which may represent a common mechanism to limit aGVHD severity in both strategies.

To assess protein synthesis capacity, we used the Click reaction with O-propargyl-puromycin (OPP), which does not require uptake by any amino acid transporter of the cells (Pelgrom et al, 2023). Genetic deletion of SLC7A5 and the absence of Leu significantly reduced the capacity of protein synthesis after activation (Fig. 4D). However, both SLC7A5ΔCD4 and SLC7A5WT T cells cultured in (-)Leu medium retained protein synthesis activity in comparison with cells treated with inhibitors. Considering the relevance of SLC7A5-mediated L-Leu uptake in the control of mTOR by activated T cells (Sinclair et al, 2013), we comparatively analyzed the expression of pS6 in Tmt+ cells at 7 days of transplant (Fig. 4E). Our results showed a significant reduction in the percentage and mean fluorescence intensity (MFI) values of pS6, i.e., in mTOR activity, in SLC7A5ΔCD4 T cells as compared to SLC7A5WT cells in mice fed with regular diet (Fig. 4E). However, the SLC7A5WT cells detected in (-)Leu diet-fed mice showed similar percentages of pS6 in CD4 and CD8 T cells as SLC7A5WT cells from regular-diet-fed mice, although the expression levels of pS6 (MFI values) were clearly diminished (Fig. 4E). Altogether, these data confirm that dietary restriction of SLC7A5-mediated L-Leu uptake by allogenic T cells is sufficient to reduce protein synthesis and the mTOR pathway, controlling expression of GzmB in CD8 T cells. However, significant differences are observed regarding the secretion of IFNγ between SLC7A5ΔCD4 T cells and SLC7A5WT T cells without L-Leu.

In addition, the frequency of FoxP3-expressing CD4 T cells was measured in the spleen, colon, skin and lung of SLC7A5ΔCD4 → BALB/c mice, and SLC7A5WT → BALB/c mice fed with regular or (-)Leu-diet, 7 days after transplant (Fig. EV4). A significant increase of FoxP3+ was detected in the Tmt+CD4+ T cells from spleen and skin of SLC7A5ΔCD4 → BALB/c mice, as compared to SLC7A5WT → BALB/c mice fed with regular diet, but not in colon and lung (Fig. EV4). Interestingly, a significantly higher percentage of FoxP3+ Tmt+CD4+ T cells was also observed in the mucosal areas (colon, skin, and lung) of SLC7A5WT → BALB/ c mice fed with (-)Leu-diet as compared to SLC7A5WT → BALB/c

mice fed with regular diet, but not in the spleen (Fig. EV4). Considering that Treg cells were not excluded from STC from donor mice during aGVHD experiments, we cannot rule out that Treg cells detected in the spleen and tissues after 7 days of transplant are the same cells that were injected. Alternatively, mTOR reduction can increase FoxP3 expression in our experimental settings (Sauer et al, 2008). Nevertheless, further functional experiments will be required to assess the relevance of Treg cells in aGVHD in the absence of SLC7A5 or L-Leu.

## GVHD modifies serum levels of metabolites from the tricarboxylic acid cycle and ketogenesis-derived from L-Leu degradation

Serum samples from control BALB/c mice fed with regular or (-)Leu diet, SLC7A5WT → BALB/c mice fed with regular or (-)Leu diet, and SLC7A5ΔCD4 → BALB/c mice were analyzed by nuclear magnetic resonance (NMR) spectroscopy. This experimental setting allowed us to reveal serum metabolites that were affected by the aGVHD, the deprivation of L-Leu and the expansion of allogenic T cells. Principal component analysis (PCA) was applied as an unsupervised multivariate statistical analysis method based on the overall metabolite profile (Fig. 5A, Table EV1). SLC7A5WT → BALB/c irradiated groups, both regular and (-)Leu diet, were differentiated between them and from control and SLC7A5ΔCD4 → BALB/c mice (Fig. 5A). Remarkably, the SLC7A5ΔCD4 → BALB/c group metabolic profile was comparable and overlapping to control groups, suggesting that blood metabolic changes observed in SLC7A5WT → BALB/c mice are mainly induced by the activation and expansion of allogenic T cells (Fig. 5A). The inflammatory markers (Glyc, GlycA, and GlycB,) were upregulated in both SLC7A5WT → BALB/c irradiated groups, regardless of diet, but not in the SLC7A5ΔCD4 → BALB/c mice (Fig. 5B,C). Importantly, the metabolomics analysis confirms the reduction of serum L-Leu levels in mice fed with an L-Leu-deprived diet, in both control and SLC7A5WT → BALB/c irradiated groups (Fig. 5B,C). Serum levels of succinic acid and citric acid were down-regulated in SLC7A5WT → BALB/c mice as compared to the control group, suggesting an increase in energy production by the tricarboxylic acid (TCA) cycle in aGVHD (Fig. 5B,C) (O'Neill et al, 2016). The metabolite 2-oxoglutarate, a key intermediate of the mitochondrial TCA cycle, was increased in SLC7A5WT → BALB/c mice fed with regular diet, as compared to the control group (Fig. 5B,C). 2-oxoglutarate is also generated by the reversible oxidation of Glu during the glutaminolysis pathway (O'Neill et al, 2016). Indeed, the AA Glu was less abundant in the serum of SLC7A5WT → BALB/c mice fed with regular diet, suggesting glutaminolysis activity (Fig. 5B,C). In addition, L-Leu is a strictly ketogenic AA that is ultimately converted

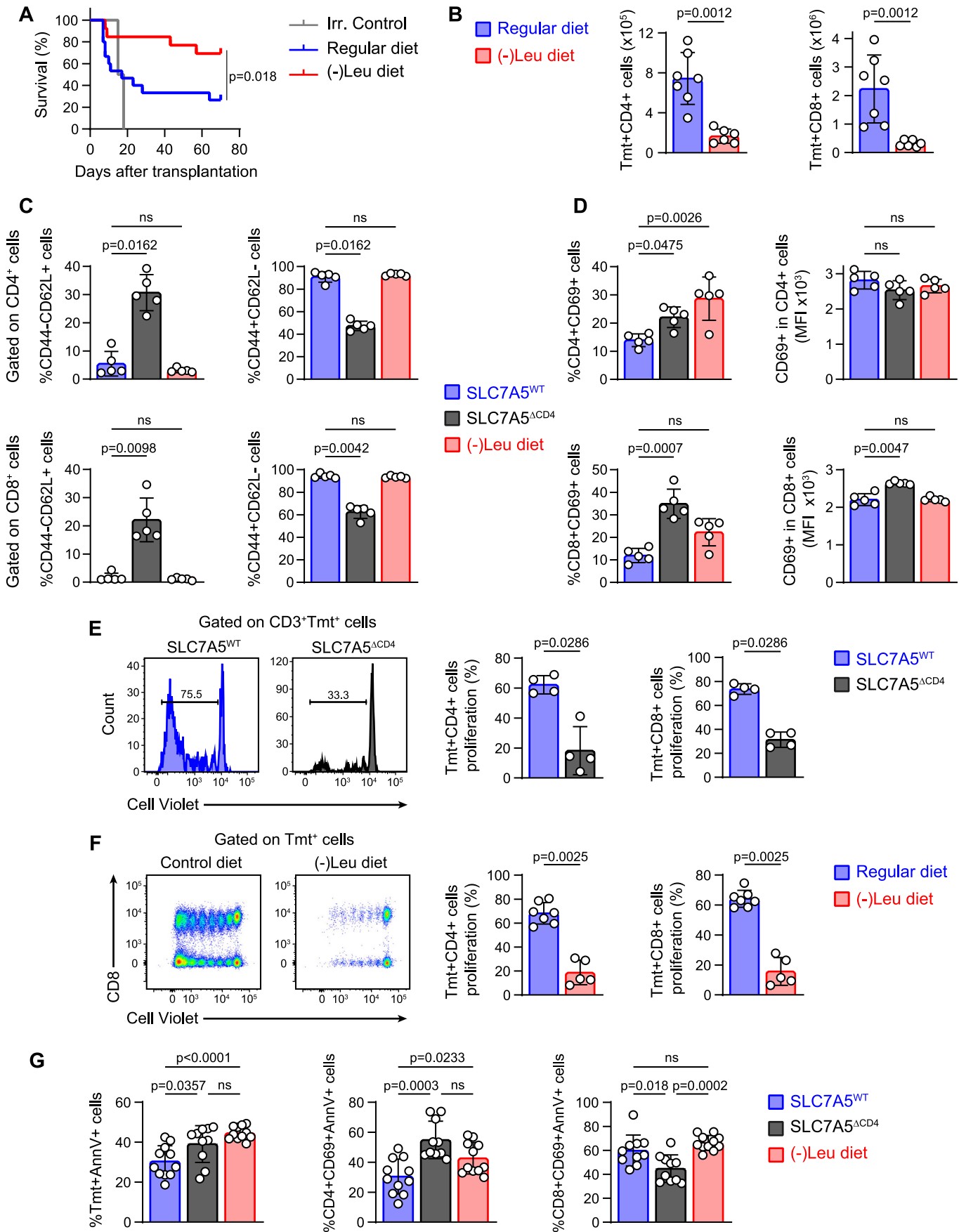

◄ **Figure 3. Dietary restriction of L-Leu controls aGVHD mortality, reduces proliferation and promotes apoptosis of allogenic T cells.**

(A) Survival curve of irradiation controls BALB/c mice (gray, $N = 2$), SLC7A5$^{WT}$ → BALB/c regular diet group (blue, $N = 15$) and (-)Leu diet group (red, $N = 13$). A pool of two independent experiments is shown. Long-rank (Mantel–Cox) and Cehan-Breslow-Wilcoxon test. (B) Absolute numbers of Tmt$^+$CD4$^+$ and Tmt$^+$CD8$^+$ T cells in the spleen of SLC7A5$^{WT}$ → BALB/c fed with regular or (-)Leu diet at 7 d post-transplant. (C) Frequencies of naive (CD44$^-$CD62L$^+$) and effector memory (CD44$^+$CD62L$^-$) T cells gated on allogenic CD4$^+$ and CD8$^+$ T cells of spleen from (-)Leu diet-fed SLC7A5$^{WT}$ → BALB/c mice and from regular-diet fed BALB/c host mice simultaneously injected with SLC7A5$^{\Delta CD4}$ and SLC7A5$^{WT}$ T cells at 7 d post-transplant. (D) Frequencies (left) and mean fluorescence intensity (MFI, right) of CD69 in allogenic CD4$^+$ and CD8$^+$ T cells from spleen of mice described in (C). (E) Representative histogram plots (left) of CellTrace Violet (CV) intensity loss gated on CD3$^+$Tmt$^+$ cells and frequencies (right) of proliferating cells in Tmt$^+$CD4$^+$ and Tmt$^+$CD8$^+$ from spleen of SLC7A5$^{WT}$ → BALB/c and SLC7A5$^{\Delta CD4}$ → BALB/c mice at 3 d post-transplant. (F) Density plots (left) of CV intensity loss gated on total Tmt$^+$SLC7A5$^{WT}$ cells and frequencies (right) of proliferating cells in Tmt$^+$CD4$^+$ and Tmt$^+$CD8$^+$ cells from the spleen of mice fed with regular or (-)Leu diet for 3 week and analyzed after 3 d of transplant. (G) Frequencies of Annexin V (AnnV) detection in total Tmt$^+$, CD4$^+$CD69$^+$ and CD8$^+$CD69$^+$ cells from mesenteric lymph nodes (mLN) of SLC7A5$^{WT}$ → BALB/c mice fed with regular or (-)Leu diet and SLC7A5$^{\Delta CD4}$ → BALB/c mice at 7 d post-transplant. (B–F): $N = 4$–7 per group, (G): $N = 10$–11 per group, mean ± SD, two-tailed Mann–Whitney's test (B, E, F, G) or Kruskal–Wallis test with Dunn's post-test (C, D), ns: not significant. Source data are available online for this figure.

to acetyl-CoA and acetoacetate, thus also contributing to the increased levels of 2-oxoglutarate through the TCA cycle, as well as to the generation of ketone bodies. The ketone bodies acetoacetate and its derived metabolite, 3-hydroxybutyric acid, were highly increased in SLC7A5$^{WT}$ → BALB/c mice fed with regular diet, whereas they were decreased in mice with Leu dietary restriction (Fig. 5B,C). Hence, the increase in 2-oxoglutarate, acetoacetate and 3-hydroxybutyric acid seems to be the main metabolic signature induced by the inflammatory response in aGVHD, and they require the presence of Leu in the diet.

## Effects of SLC7A5-genetic deletion and L-Leu deprivation on glycolysis and mitochondrial T cell metabolism

The role of Leu-depletion in T cell metabolism after activation has not been so far explored. Hence, Seahorse assays were performed to ascertain the glycolytic and oxidative phosphorylation capacities of SLC7A5$^{\Delta CD4}$ and SLC7A5$^{WT}$ CD4$^+$ T cells activated either in complete or (-)Leu medium. The genetic deletion of SLC7A5 in activated CD4$^+$ T cells severely affected glycolysis, involving decreased glycolytic capacity and ATP production (Fig. 6A). It also affected mitochondrial activity when using glucose as an energy source, with decreased basal and maximal mitochondrial respiratory capacity, proton leak and coupling efficiency, as compared to SLC7A5$^{WT}$ cells cultured in control medium (Fig. 6A). SLC7A5$^{WT}$ CD4$^+$ T cells activated without L-Leu showed a significant reduction in glycolytic parameters, including glycolysis and glycolytic capacity, although no alterations in non-glycolytic acidification of the media were observed (Fig. 6B). However, mitochondrial function was mainly preserved, with no changes in ATP levels and mitochondrial basal and maximal respiration (Fig. 6B). Indeed, Leu depletion reduced proton leak while increased coupling efficiency of SLC7A5$^{WT}$ cells, which was not observed in SLC7A5$^{\Delta CD4}$ CD4$^+$ T cells (Fig. 6A,B). Since mTORC1 signaling regulates HIF1α expression and glycolysis (Finlay et al, 2012; Shi et al, 2011), we also explored the effect of mTORC1 inhibition by rapamycin in SLC7A5$^{WT}$ CD4$^+$ T cells in vitro activated with or without Leu. The effect of rapamycin-induced inhibition of glycolysis was milder than the effect of Leu-deprivation and was not observed in the absence of Leu (Fig. 6B). These results suggest that the strong effect of Leu deprivation on glycolysis of activated CD4$^+$ T cells is not mediated by the reduction of mTORC1 signaling. Importantly, Leu deprivation and mTORC1 inhibition did not alter mitochondrial function and ATP production, as was observed in SLC7A5$^{\Delta CD4}$ T cells.

## Dietary restriction of Leu but not SLC7A5-genetic deletion controls tumor growth and maintains GVT response

We assessed whether genetic deletion of SLC7A5 in T cells affect GVT response using a BALB/c compatible murine lung carcinoma cell line, KLN-205, injected in the flank of recipient mice (Fig. EV5). Neither control nor SLC7A5$^{\Delta CD4}$ → BALB/c mice with tumors died, despite increased tumor growth (Fig. 7A). Hence, the mortality mainly associated with GVHD was higher in SLC7A5$^{WT}$ → BALB/c mice injected with tumors than in SLC7A5$^{\Delta CD4}$ → BALB/c with tumors (Fig. 7A). Only those mice that survived during 25 days of assay were finally included in the tumor volume and tumor growth graphics. As observed by tumor volume and growth data, GVT response mediated by Tmt$^+$SLC7A5$^{WT}$ T cells was responsible for the significant reduction of tumor growth compared to control BALB/c mice from day 17 to day 25 of assay (Fig. 7B). However, tumors grew similarly in SLC7A5$^{\Delta CD4}$ → BALB/c and control BALB/c mice (Fig. 7B), indicating the absence of GVT response when allogenic T cells were depleted of SLC7A5.

The relative percentage and total numbers of CD4 and CD8 Tmt$^+$ T cells in the spleen were analyzed after 25 days of tumor injection (Fig. 7C). Tmt$^+$CD8$^+$ SLC7A5$^{WT}$ T cells were expanded in comparison with CD4$^+$ T cells, but this effect was not observed with Tmt$^+$SLC7A5$^{\Delta CD4}$ T cells (Fig. 7C). Regarding activation and memory markers, the fraction of effector/memory T cells (CD44$^+$CD62L$^-$) in the spleen was significantly increased in Tmt$^+$SLC7A5$^{WT}$ cells, while SLC7A5$^{\Delta CD4}$ T cells showed an increased fraction of naive T cells (CD62L$^+$CD44$^-$) and central memory T cells (CD44$^+$CD62L$^+$) exclusively for CD4$^+$ T cells (Fig. 7D). We also tested for exhaustion markers PD1 and TIM3 (Fig. 7E). Co-expression of TIM3 and PD1 identifies a more severe T-cell exhaustion phenotype in tumor infiltrating lymphocytes (Roussel et al, 2021). SLC7A5$^{\Delta CD4}$ cells showed a reduced percentage of PD1$^+$TIM3$^+$ and of total PD1$^+$ in the CD44$^+$CD62L$^-$ fraction of Tmt$^+$CD4$^+$ and Tmt$^+$CD8$^+$ cells (Fig. 7E). Despite the reduced expression of exhaustion markers, SLC7A5$^{\Delta CD4}$ T cells fail to generate GVT response.

We also explored the effect of L-Leu-deprived diet on tumor growth in control syngenic BALB/c mice and in mice with allogenic Tmt$^+$SLC7A5$^{WT}$ cells (Fig. EV6A). Control BALB/c mice fed with either regular or (-)Leu diet survived the 25 days of assay and did not die because of tumor injection (Fig. 8A). The mortality of mice with GVHD and tumor was significantly reduced in mice fed with

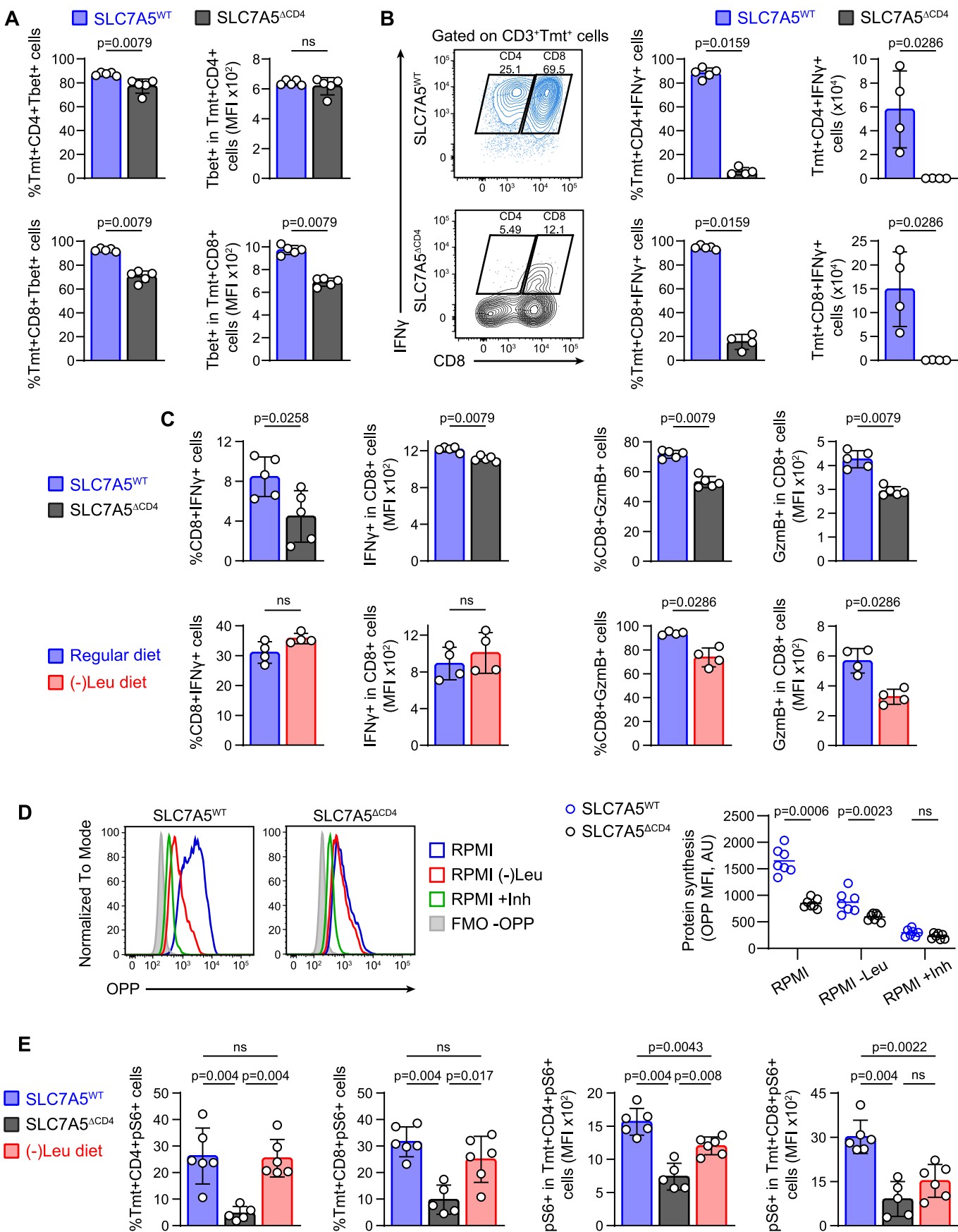

◀

Figure 4.   Leu deprivation and SLC7A5 deletion regulate cytokine secretion, blunt protein synthesis and dampen mTORC1 pathway.

(A) Frequencies (left) and mean fluorescence intensity (MFI, right) of Tbet in Tmt⁺CD4⁺ (top) and Tmt⁺CD8⁺ (bottom) cells from the spleen of SLC7A5$^{WT}$ → BALB/c and SLC7A5$^{\Delta CD4}$ → BALB/c mice at 7 d post-transplant. (B) Cells from mesenteric lymph nodes (mLN) of SLC7A5$^{WT}$ → BALB/c and SLC7A5$^{\Delta CD4}$ → BALB/c mice were re-activated in vitro ON with anti-CD3/anti-CD28 and later incubated with PMA/ionomycin/monensin for 4 h. Representative density plots (left) of IFNγ gated on CD3⁺Tmt⁺ cells from mLN are shown. Frequencies (middle) and absolute numbers (right) of IFNγ in Tmt⁺CD4⁺ (top) and Tmt⁺CD8⁺ (bottom) cells. (C) Frequency and MFI of IFNγ (left) and granzyme B (GzmB) (right) in CD8⁺ T cells of spleen and mLN cell suspensions from SLC7A5$^{WT}$ and SLC7A5$^{\Delta CD4}$ mice activated in vitro with concanavalin A for 48 h plus PMA/ionomycin/brefeldin for 3 h in complete or (-)Leu-medium. (D) Representative histogram plots (left) and MFI of OPP intensity (right) gated on live cells in purified CD8⁺ T cells from spleen and mLN of SLC7A5$^{WT}$ and SLC7A5$^{\Delta CD4}$ mice stimulated ON with anti-CD3/anti-CD28 and later incubated with PMA/ionomycin for 3 h in complete or (-)Leu- medium. A mix of puromycin and cycloheximide inhibitors was used as control. (E) Frequencies and MFI of pS6 in Tmt⁺CD4⁺ and Tmt⁺CD8⁺ T cells from the spleen of SLC7A5$^{WT}$ → BALB/c fed with regular or (-)Leu diet and SLC7A5$^{\Delta CD4}$ → BALB/c mice at 7 d post-transplant and stimulated ON with anti-CD3/anti-CD28 in complete or (-)Leu medium. (A–E): N = 4–7 per group, mean ± SD, two-tailed Mann–Whitney's test, ns: not significant. Source data are available online for this figure.

(-)Leu diet compared to regular diet (Fig. 8A). Again, only mice that survived during the 25 days were included in the graphics of tumor volume (Figs. 8B,C and EV6B). The effect of GVT was clearly observed in SLC7A5$^{WT}$ → BALB/c mice fed with regular diet, in which a significant reduction of tumor volume was detected compared to regular diet control BALB/c mice (Fig. 8B). In addition, the L-Leu-deficient diet significantly reduced the tumor volume, as compared between syngenic control BALB/c groups (Fig. EV6B). Interestingly, when mice fed with (-)L-Leu diet were compared, a significant reduction of tumor volume was detected in SLC7A5$^{WT}$ → BALB/c mice compared to control BALB/c mice, suggesting an additional effect of GVT response even in the absence of L-Leu (Fig. 8C).

After 25 days of tumor injection, all mice were sacrificed to analyze the immune response in the tumor and in the spleen. The hematoxylin/eosin-stained sections confirmed the reduction of tumor volume and the presence of increased necrotic area in mice transplanted with allogenic T cells, in comparison with tumors injected in control BALB/c mice (Fig. 8D). Moreover, we confirmed the increase of CD3⁺ lymphocyte infiltration in tumors from mice with GVHD in the regular diet, as compared to control BALB/c mice (Fig. 8E). Importantly, infiltrating CD3⁺ T cells were also detected in tumors of SLC7A5$^{WT}$ → BALB/c mice fed with (-)Leu diet. This result confirmed the presence of T-cell-mediated tumor response (Fig. 8E), despite the lower number of Tmt⁺CD4⁺ and Tmt⁺CD8⁺ T cells detected in the spleen of these mice compared to SLC7A5$^{WT}$ → BALB/c fed with regular diet (Fig. 8F). We next compared the expression of CD44 and CD62L of Tmt⁺SLC7A5$^{WT}$ cells in GVHD groups, fed with regular or (-)Leu diet, with CD4 and CD8 T cells from their respective control BALB/c mice (Figs. 8G and EV6C,D). Allogenic T cells from both GVHD groups showed an increased fraction of effector memory cells (CD44⁺CD62L⁻) and a decreased proportion of naive T cells (CD44⁻CD62L⁺), as compared to CD4 and CD8 T cells detected in BALB/c control groups (Figs. 8G and EV6C,D). Remarkably, the induction of central memory T cells (CD44⁺CD62L⁺) was increased in CD8⁺ T cells in both GVHD or control groups without L-Leu, suggesting a diet-associated effect in the generation of this population (Figs. 8G and EV6C). We also analyzed the expression of PD1 and TIM3 exhaustion markers in CD44⁺ T cells. No significant differences were detected in the percentage of total PD1⁺ and PD1⁺TIM3⁺ between GVHD groups fed with regular or (-)Leu diet (Fig. EV6E). These results suggest that Leu deficiency does not increase the exhaustion status of allogenic T cells, according to the expression of PD1 and TIM3 markers. Finally, we explored the expression of the transcription factor T-cell factor 1 (TCF1) in the

CD44⁺ fraction of CD4 and CD8 T cells as indicative of their stem-like properties (Siddiqui et al, 2019) (Fig. 8H). Although we did not find differences between GVHD groups in the percentage of CD44⁺CD4⁺ and CD44⁺CD8⁺ T cells expressing TCF1, we found that the levels of expression of this transcriptional factor are upregulated in both groups of mice fed without Leu, suggesting the induction of a stemness phenotype by Leu deprivation.

## Discussion

GVHD therapies must comply with the requirements of reducing the risk of allogenic T-cell-induced target tissue injury while maintaining allogenic T cell activity against tumors. Therefore, therapies that successfully control allogenic T cell proliferation and activation as well as prevent tumor growth are desirable. Importantly, SLC7A5 molecule expression is upregulated in different hematological and solid cancer types where it plays an essential role in L-Leu uptake and mTORC1 activation and has been associated with poor outcomes (Ichinoe et al, 2021; Jigjidkhorloo et al, 2021; Rosilio et al, 2015; Yanagisawa et al, 2014). Thus, we considered that the strategy of SLC7A5 inhibition by L-Leu deprivation potentially fulfills the requirement of controlling GVHD onset and preventing cancer expansion.

Our results demonstrate that targeting SLC7A5-mediated L-Leu uptake by genetic deletion of the amino acid transporter or by reduced L-Leu intake from the diet efficiently controls allogenic T-cell expansion and mortality in aGVHD model. The protective effect of SLC7A5-deletion in T cells has been previously described in inflammatory mouse models of psoriasis (Cibrian et al, 2020), allergic diseases (Hayashi et al, 2020), and rheumatoid arthritis (Ogbechi et al, 2023). Few studies on nutritional interventions in GVHD have been reported to date. Importantly, it has been described that the metabolism of branched-chain amino acids (BCAAs) (Leu, Ile, Val) is increased in patients with aGVHD, and altered pre-transplant metabolism of BCAAs has been associated with an increased risk of aGVHD (Reikvam et al, 2016). As far as we know, this is the first study to report reduced mortality in a model of aGVHD in mice with exclusive L-Leu restriction in the diet.

In addition, the genetic deletion of SLC7A5 and the absence of L-Leu in the diet increase allogenic CD4 but not CD8 T cell apoptosis in vivo. The induction of apoptosis by the inhibition of SLC7A5 or dietary L-Leu restriction is a mechanism that might contribute to control CD4 T cell-mediated immune response which has not been described previously, although this effect has been well

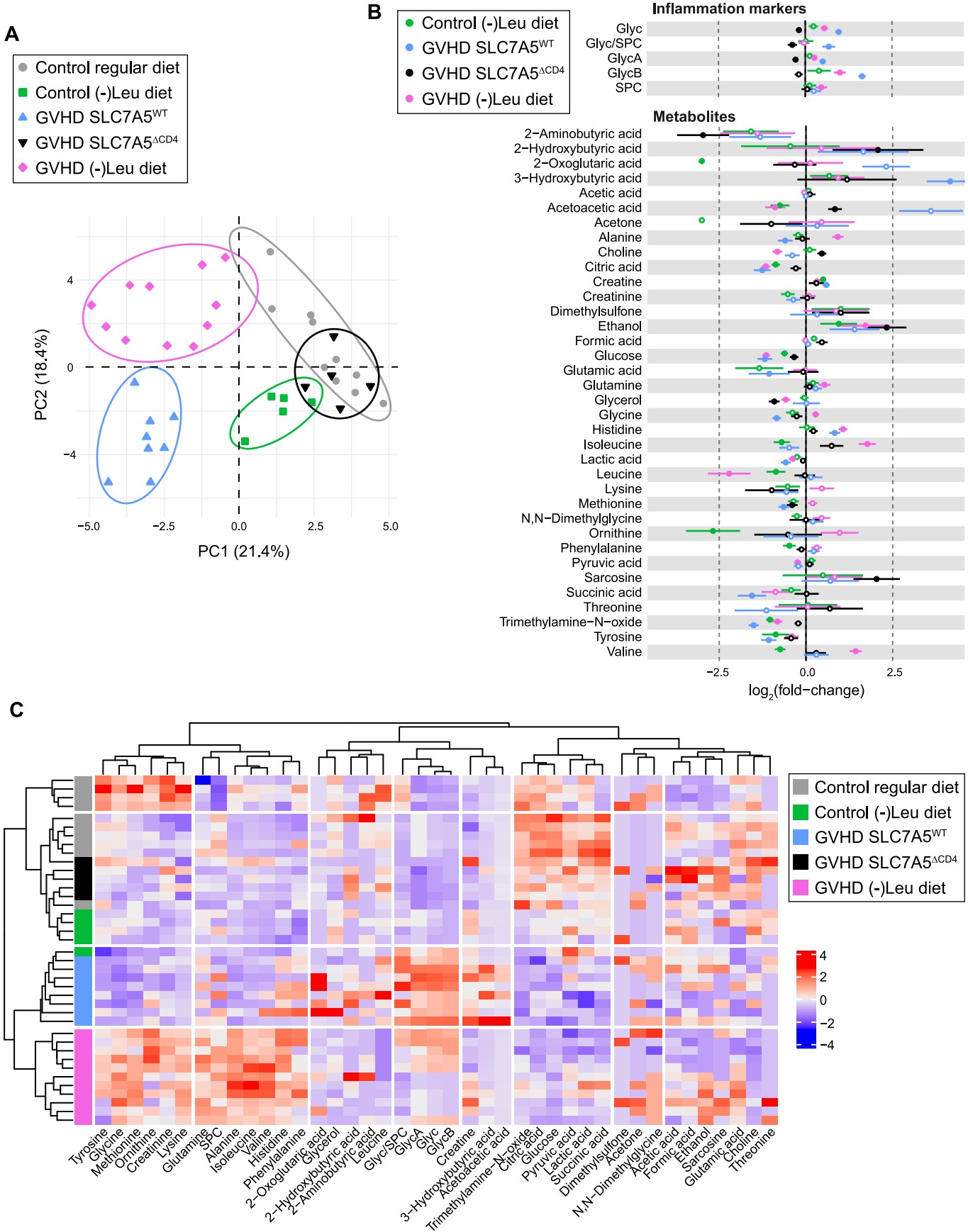

**Figure 5. Metabolomic changes induced by SLC7A5 deficiency and Leu deprivation in aGVHD.**

Serum of control regular diet mice (N = 10) and control (-)Leu diet mice (N = 5) (non-irradiated and non-transplanted, gray and green, respectively), SLC7A5^WT → BALB/c mice fed with regular diet (blue, N = 8), SLC7A5^WT → BALB/c fed with (-)Leu diet (pink, N = 11) and SLC7A5^ΔCD4 → BALB/c mice (black, N = 5) was collected 7 days post-transplant and metabolites were analyzed by nuclear magnetic resonance (NMR). (A) Unsupervised multivariate analysis via Principal Component Analysis (PCA). (B) Forest plot of serum metabolomics of all groups in reference to regular diet group. X axis represents the logarithm in base 2 of fold change (log2FC) of each metabolite respect to the reference group. Horizontal lines indicate standard errors and filled circles represent statistically significant differences (p < 0.05) obtained through a Student's t-test. (C) Heatmap of 40 metabolites analyzed. The figure shows the clustering results in the form of a dendrogram. Relative abundance is represented along a color gradient from blue (diminished) to red (increased). GlycA α-1-acid glycoprotein N-acetyl-glucosamine (N-acetyl) signal A, GlycB α-1-acid glycoprotein N-acetyl-glucosamine (N-acetyl) signal B, SPC supramolecular phospholipids composite peak. Source data are available online for this figure.

documented for cancer cells (Choi et al, 2017; Rosilio et al, 2015; Sheen et al, 2011; Yothaisong et al, 2017). The lack of increased apoptosis in SLC7A5 or L-Leu-depleted CD8 T cells is most likely due to reduced expression of granzyme B (GzmB) (Laforge et al, 2006), which, together with reduced proliferation, control aGVHD.

A defect in pS6 phosphorylation in SLC7A5^ΔCD4 T cells in aGVHD model was detected, which agrees with the previously described role of SLC7A5 on the mTORC1 pathway (Sinclair et al, 2013). Furthermore, the absence of L-Leu dampened mTOR signaling in allogenic SLC7A5^WT T cells, but the effect was milder than by genetic deletion of SLC7A5. Three Leu-sensing mechanisms regulating mTORC1 activity have been uncovered. Two of them involve cytoplasmic Leu sensors: the tRNA-charging enzyme leucyl-tRNA synthetase (Han et al, 2012) and the protein Sestrin2 (Wolfson et al, 2016). The third mechanism relies on the lysosomal L-Leu sensing, mediated by SLC7A5 localization in the lysosomal membrane (Milkereit et al, 2015). All these mechanisms would be affected by SLC7A5 genetic deletion. In contrast, a potential increase of autophagy could account for lysosomal accumulation of L-Leu and residual activation of mTORC1 by the expression of SLC7A5 in the lysosome of SLC7A5^WT T cells activated without L-Leu, similarly to what is observed in cells deficient for SLC3A2 (Cormerais et al, 2016). In addition, other amino acids (AA) such as Arg or Ile could be activating mTORC1 in the absence of Leu (Ikeda et al, 2017; Shi et al, 2019), but further studies are required to clarify these differences. Importantly, our data also demonstrate a significant reduction of protein synthesis in SLC7A5^ΔCD4 T cells, as well as in SLC7A5^WT T cells activated in the absence of L-Leu, which impacts on T cell proliferation and cytokine synthesis. In addition, we showed that both SLC7A5-deletion and L-Leu-deprivation reduce GzmB expression in CD8+ T cells. This effect is relevant for the observed protection against aGVHD, as it has been proved that genetic deletion of GzmB is sufficient to control GVHD (Graubert et al, 1996) while contributing to maintaining GVT response (Bian et al, 2013).

Remarkably, significant differences have been detected between SLC7A5-deficient T cells and T cells activated in Leu-free conditions, both in vitro or in vivo, in GVHD or GVT models. First, regarding lymphocyte activation, SLC7A5-deletion impairs the acquisition of the effector/memory phenotype in vivo, which was not observed in the absence of L-Leu, although in both cases activated T cells expressed higher levels of CD69. We have previously described that CD69 associates with SLC7A5-SLC3A2 complex (Cibrian et al, 2016). Hence, it is conceivable that the upregulation of CD69 is induced as a compensatory mechanism by the signaling pathway activated by nutrient deprivation or mTORC1 reduction, but further experimentation will be required

to analyze this effect. Secondly, genetic deletion of SLC7A5 in T cells controls the IFNγ expression in CD4 and CD8 T cells. This effect was not observed in SLC7A5^WT T cells activated in medium without L-Leu, despite the reduction of protein synthesis capacity. The role of SLC7A5 in T cell differentiation towards Th1, Th2, and Th17 has been previously described in vitro and in vivo in mouse models of viral infection (Hayashi et al, 2013; Sinclair et al, 2013), psoriasis (Cibrian et al, 2020), allergic diseases (Hayashi et al, 2020), and rheumatoid arthritis (Ogbechi et al, 2023). However, the effect of L-Leu removal has not been explored until recently, where a similar disconnection between L-Leu deprivation and the maintenance of IFNγ secretion was observed in CD4+T cells activated in vitro (Kang et al, 2024). Finally, our seahorse data highlight a main difference in T cell metabolism between SLC7A5^ΔCD4 T cells and SLC7A5^WT T cells activated in the absence of L-Leu. The genetic deletion of SLC7A5 drastically affected glycolysis and mitochondrial respiration, causing a severe reduction in the cellular ATP pool. However, cells that were activated in the absence of L-Leu showed a marked defect in glycolysis but maintained mitochondrial respiration and ATP content. Despite the relevance of glycolysis upregulation for IFNγ expression (Siska and Rathmell, 2016), our data underscore the potential to reduce glycolysis and maintain IFNγ secretion by reducing L-Leu uptake through SL7CA5 in CD8 T cells. Similar results have been obtained with the inhibition of BCAT enzymes in in vitro activated CD4 T cells where reduced expression of HIF1α was observed and IFNγ secretion was not affected (Kang et al, 2024).

Our metabolomic analyses indicated an important elevation of 2-oxoglutarate in serum of mice with aGVHD, which was prevented by dietary restriction of L-Leu or transplantation of allogenic SLC7A5-deficient T cells. This intermediate metabolite of the TCA cycle has been associated with GVHD onset in patients (Subburaj et al, 2022). Furthermore, our metabolomic data showed an important increase in the ketone bodies acetoacetate and β-hydroxybutyrate during the onset of aGVHD in mice. The control of ketogenesis in mice with dietary restriction of L-Leu or receiving SLC7A5-deficient T cells highlights the relevance of L-Leu uptake by allogenic T cells in the generation of these metabolites. Our data suggest that L-Leu is the preferential substrate oxidized to ketone bodies during aGVHD, in addition to the potential role of fatty acid catabolism. Both acetoacetate and β-hydroxybutyrate are preferentially consumed by activated CD8+ T cells in vivo, rather than glucose, to fuel the TCA cycle through the process of ketolysis in a murine model of infection (Luda et al, 2023). Moreover, it has been recently demonstrated that β-hydroxybutyrate induces the expression of HIF1α in activated T cells, upregulating glycolysis (Kang et al, 2024). These results suggest that impaired glycolysis in cells

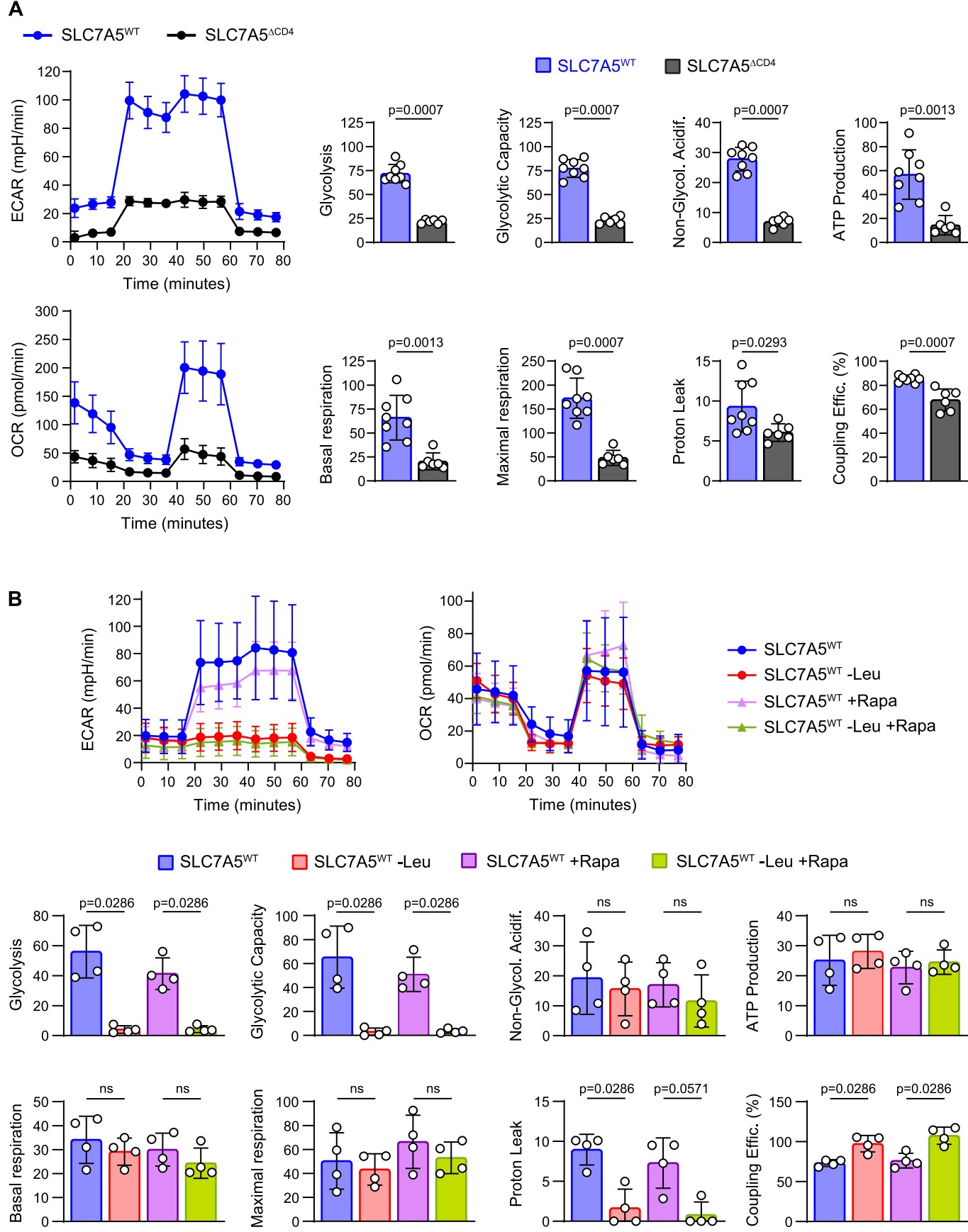

◀ **Figure 6. Impact of SLC7A5 deficiency and Leu deprivation in energetic metabolism of T cells.**

(A, B) Estimation of respiratory and glycolytic function of SLC7A5$^{WT}$ and SLC7A5$^{\Delta CD4}$ CD4$^+$ T cells activated in vitro with anti-CD3/anti-CD28 for 48 h (A) in complete or (-)Leu-medium (B). Left (A) or top (B) panels show representative profiles of extracellular acidification rate (ECAR) and oxygen consumption rate (OCR) determined by Seahorse analysis and right (A) or bottom (B) panels show parameters of glycolytic (in mpH/min) and mitochondrial (in pmol/min) function. Non-Glycol. Acidif: non-glycolytic acidification; Coupling Effic: coupling efficiency. A representative experiment of two is shown. (A): $N = 6{-}8$ per group, (B): $N = 4$ per group, mean ± SD, two-tailed Mann–Whitney's test, ns: not significant. Source data are available online for this figure.

activated without L-Leu is due to the defect of β-hydroxybutyrate synthesis in activated T cells, as not regulatory effect was observed with mTORC1 inhibitor. The relevance of β-hydroxybutyrate in GVHD patients has not yet been studied in detail, but our data suggest that the circulating level of L-Leu is a key regulator of the TCA cycle metabolite 2-oxoglutarate and β-hydroxybutyrate formation, which could significantly impact GVHD. Therefore, controlling L-Leu uptake by modifying AA levels in the diet of patients may represent a beneficial strategy for the treatment of GVHD.

Our results highlight for the first time that SLC7A5-deficient allogenic T cells fail to reject tumor due to impairment in activation, proliferation and GzmB and IFNγ expression. Importantly, our results underscore that KLN205 tumor cells depend on L-Leu for growth and survival. Further experiments will be required to explore the relevance of autophagy in the adaptation to L-Leu deprivation, as has been described for melanoma cells (Sheen et al, 2011). Besides the effect of L-Leu removal in the restriction of KLN205 tumor growth, a substantial GVT response was observed, which might be explained by the activation and preservation of IFNγ response by allogenic CD8$^+$ T cells. In addition, the defect of GzmB expression in L-Leu-depleted activated CD8 T cells seems to preserves their viability and contributes to GVT response (Bian et al, 2013). Moreover, the selective maintenance of mitochondrial metabolism in the absence of L-Leu could be responsible for the increase in the central memory compartment in allogenic T cells (Buck et al, 2016), where the role of fatty acid oxidation may be also contributing to the GVT response (Dumauthioz et al, 2021). Indeed, central memory CD8$^+$ T cells confer superior antitumor immunity compared with effector memory T cells (Klebanoff et al, 2005), and could be contributing to the GVT tumor response observed in the absence of L-Leu. Interestingly, TCF1 expression was upregulated in the absence of L-Leu, which is indicative of an increase in stem-like phenotype in activated memory T cells (Siddiqui et al, 2019). The priority of mitochondrial metabolism versus glycolysis (Wu et al, 2023) and the negative regulation of mTORC1 signaling (Karmaus et al, 2019) have been associated with the upregulation of TCF1 expression, a mechanism that might be targeted in our results with L-Leu dietary restriction.

Overall, our study highlights the reduced dietary L-Leu intake as a promising therapeutic strategy to control SLC7A5-mediated T cell expansion and GVHD and reduce tumor growth while preserving the GVT response. Additionally, L-Leu deprivation limits the systemic generation of metabolites derived from L-Leu catabolism, such as 3-hydroxybutyric acid, which could enhance T cell-mediated inflammation by exacerbating mTORC1 activation and glycolysis of immune cells.

## Methods

### Reagents and tools table

| Reagent/Resource | Reference or Source | Identifier or Catalog Number |
|---|---|---|
| **Experimental models** | | |
| KLN205 cells (*M. musculus*) | ATCC | CRL-1453 |
| BALB/cJRj (*M. musculus*) | Janvier Laboratories | |
| C57BL/6 CD45.1$^+$ (*M. musculus*) | Charles River Laboratories | B6.SJL-*Ptprc$^a$ Pepc$^b$*/BoyJ Strain #002014 |
| C57BL/6 β-actin-eGFP (*M. musculus*) | The Jackson Laboratory | C57BL/6-Tg(CAG-EGFP)131Osb/LeySopJ Strain #006567 |
| C57BL/6 CD4-Cre (*M. musculus*) | The Jackson Laboratory | B6.Cg-Tg(Cd4-Cre)1Cwi/BfluJ Strain #022071 |
| C57BL/6 *Slc7a5*$^{fl/fl}$ (*M. musculus*) | Provided by Peter M. Taylor, Dundee University, England (Poncet et al, 2014) | N/A |
| C57BL/6 Tomato reporter (*M. musculus*) | The Jackson Laboratory | B6.Cg-*Gt(ROSA)26Sor$^{tm14(CAG-tdTomato)Hze}$*/J Strain #007914 |
| CD4-Cre$^{+/-}$ *Slc7a5*$^{fl/fl}$ Tmt$^{fl/wt}$ (*M. musculus*) | In-house sourced, CNIC animal facility (Cibrian et al, 2020) | N/A |
| **Antibodies** | | |
| CD3ε, clone 145-2C11 (for activation) | Tonbo Biosciences | 70-0031-M001 |
| CD3ε rabbit polyclonal (for IF) | Dako | A0452 |
| CD3ε-biotin, clone 145-2C11 (for depletion) | BD Biosciences | 553060 |
| CD3ε-BV421, clone 145-2C11 | BD Biosciences | 562600 |
| CD3ε-FITC, clone 145-2C11 | BD Biosciences | 553062 |
| CD3ε-APC-Cy7, clone 145-2C11 | BioLegend | 100330 |
| CD3ε-APC, clone 145-2C11 | BD Biosciences | 553066 |
| CD3ε-AF647, clone 17A2 | BD Biosciences | 557869 |
| CD4-BV421, clone RM4-5 | BD Biosciences | 740007 |

| Reagent/Resource | Reference or Source | Identifier or Catalog Number |
| --- | --- | --- |
| CD4-APC, clone RM4-5 | Tonbo Biosciences | 20-0042-U100 |
| CD4-PerCP, clone RM4-5 | BD Biosciences | 553052 |
| CD4-PE-Cy7, clone RM4-5 | Tonbo Biosciences | 60-0041-U100 |
| CD8α, clone 4SM15 (for IF) | ThermoFisher | 14-0808-82 |
| CD8α-FITC, clone 53-6.7 | BD Biosciences | 553030 |
| CD8α-APC, clone 53-6.7 | BD Biosciences | 553035 |
| CD8α-APC-Fire 750, clone 53-6.7 | BioLegend | 100766 |
| CD8α-PE-Cy7, clone 53-6.7 | Tonbo Biosciences | 60-0081-U100 |
| CD11b-biotin, clone M1/70 (for depletion) | BD Biosciences | 553309 |
| CD11b-vF450, clone M1/70 | Tonbo Biosciences | 75-0112-U100 |
| CD11b-FITC, clone M1/70 | Tonbo Biosciences | 35-0112-U500 |
| CD11c-biotin, clone HL3 (for depletion) | BD Biosciences | 553800 |
| CD16/CD32 (FcBlock), clone 2.4G2 | Tonbo Biosciences | 70-0161-M001 |
| CD19-biotin, clone 1D3 (for depletion) | Tonbo Biosciences | 30-0193-U500 |
| CD28, clone 37.51 | Tonbo Biosciences | 70-0281-U500 |
| CD44-vF450, clone IM7 | Tonbo Biosciences | 75-0441-U100 |
| CD44-FITC, clone IM7 | Tonbo Biosciences | 35-0441-U500 |
| CD45.1-V450, clone A20 | BD Biosciences | 560520 |
| CD45.1-PE-Cy7, clone A20 | Tonbo Biosciences | 60-0453-U100 |
| CD45.2-vF450, clone 104 | Tonbo Biosciences | 75-0454-U100 |
| CD45.2-V450, clone 104 | BD Biosciences | 560697 |
| CD45.2-APC, clone 104 | eBiosciences | 17-0454-81 |
| CD45.2-APC-Cy7, clone 104 | BioLegend | 109824 |
| CD45.2-PE-Cy7, clone 104 | Tonbo Biosciences | 60-0454-U100 |
| CD45R (B220)-biotin, clone RA3-6B2 (for depletion) | Tonbo Biosciences | 30-0452-U500 |
| CD49b-biotin, clone DX5 (for depletion) | BD Biosciences | 553856 |
| CD62L-APC, clone MEL-14 | Tonbo Biosciences | 20-0621-U100 |
| CD69-BV421, clone H1.2F3 | BD Biosciences | 562920 |

| Reagent/Resource | Reference or Source | Identifier or Catalog Number |
| --- | --- | --- |
| CD69-PE-Cy7, clone H1.2F3 | Tonbo Biosciences | 60-0691-U100 |
| CD69-APC-Cy7, clone H1.2F3 | BD Biosciences | 561240 |
| CD69- PerCP-Cy5.5, clone H1.2F3 | BD Biosciences | 561931 |
| FoxP3-APC, clone FJK-16s | Invitrogen | 17-5773-80 |
| GFP chicken polyclonal (for IF) | Abcam | ab13970 |
| Granzyme B-PE-Cy7, clone NGZB | eBioscience | 25-8898-82 |
| IFNγ-APC, clone XMG1.2 | Tonbo Biosciences | 20-7311-U100 |
| I-A/I-E (MHC-II)-biotin, clone 2G9 (for depletion) | BD Biosciences | 553622 |
| Ly6C-PE-Cy7, clone HK1.4 | BioLegend | 128018 |
| Ly6G-PerCP, clone 1A8 | Tonbo Biosciences | 65-1276-U100 |
| PD1 (CD279)-BV421, clone 29F.1A12 | BioLegend | 135217 |
| Tbet-PerCP-Cy5.5, clone eBio4B10 | Invitrogen | 45-5825-82 |
| TCF1-BV421, clone S33-966 | BD Biosciences | 566692 |
| TCRγδ-PerCP-Cy5.5, clone GL3 | BioLegend | 118118 |
| tdTomato goat polyclonal (for IF) | Quimigen | AB8181-200 |
| TER-119-biotin, clone TER-119 (for depletion) | Tonbo Biosciences | 30-5921-U500 |
| TIM3 (CD366)-PE-Cy7, clone RMT3-23 | BioLegend | 119715 |
| Donkey α-chicken-AF647 (for IF) | Invitrogen | A78952 |
| Donkey α-goat-AF647 (for IF) | Invitrogen | A32849 |
| Donkey α-goat-AF568 (for IF) | Invitrogen | A-11057 |
| Chicken α-rabbit-AF647 | Invitrogen | |
| Donkey α-rabbit-AF555 (for IF) | Invitrogen | A-31572 |
| Donkey α-rat-AF488 (for IF) | Invitrogen | A-21208 |
| **Chemicals, Enzymes and other reagents** | | |
| ACK lysis buffer | Lonza | 10-548E |
| Amino acid rodent diet (regular diet) | Research Diets, Inc | A10021B |
| Amino acid rodent diet without added leucine (-Leu diet) | Research Diets, Inc | A05080202i |

| Reagent/Resource | Reference or Source | Identifier or Catalog Number |
|---|---|---|
| Annexin V-CF Blue 7-AAD Apoptosis Staining/Detection Kit | Abcam | ab214663 |
| Antimycin A | Sigma-Aldrich | A8674-50MG |
| BAM15 | Sigma-Aldrich | SML1760-5MG |
| Brefeldin A | Sigma-Aldrich | B7651-5MG |
| CellTrace Violet Cell Proliferation Kit | Invitrogen | C34557 |
| Click-iT Plus OPP-AF488 Protein Synthesis Assay Kits | Life technologies | C10456 |
| Collagenase IV | Sigma-Aldrich | C5138-500MG |
| Concanavalin A type IV-S | Sigma-Aldrich | C2010-100mg |
| Cycloheximide | Sigma-Aldrich | C1988-1G |
| Cytofix/cytoper fixation/permeabilization kit | BD Biosciences | 554714 |
| 2-deoxy-D-glucose | Sigma-Aldrich | D8375-10MG |
| DNase I | Roche | 10104159001 |
| Dulbecco's phosphate buffered saline (dPBS) 10X | Biowest | X0515 |
| EnVision FLEX DAB+ Substrate Chromogen System (Dako Omnis) | Agilent | GV82511-2 |
| EasySep Mouse Streptavidin RapidSphere Isolation Kit | STEMCELL Technologies | 19860 |
| Fetal bovine serum (FBS) | Sigma | F7524 |
| Formalin solution, neutral buffered, 10% | Sigma-Aldrich | HT501320-9.5L |
| FoxP3/Transcription Factor Staining Buffer Set | eBioscience | 00-5523-00 |
| L-Gln | Lonza | 17-605E |
| GolgiStop (monensin) | BD Biosciences | 554724 |
| Hanks' Balanced Salt solution (HBSS) 1X | Lonza | BE10-547F |
| HEPES | HyClone | SH30237.01 |
| Ionomycin | Sigma-Aldrich | I0634 |
| Liberase TL | Roche | 5401020001 |
| Liberase TM Research Grade | Roche | 5401127001 |
| LIVE/DEAD Fixable Yellow Dead Cell Stain | Invitrogen | L34968 |
| MEM Medium | Gibco | 11095-080 |
| Na-pyruvate | HyClone | SH30239.01 |
| NEAA | HyClone | SH30238.01 |
| Oligomycin A | Sigma-Aldrich | 75351-5MG |

| Reagent/Resource | Reference or Source | Identifier or Catalog Number |
|---|---|---|
| Pancoll human, density: 1.077 g/mL (Ficoll separation solution) | PAN-Biotech | P04-60500 |
| Paraformaldehyde (PFA) | Electron Microscopy Sc. | 15710 |
| Penicillin-streptomycin | Gibco | 15140-122 |
| Percoll, density: 1.129 g/mL | GE Healthcare | 17-0891-02 |
| Phorbol 12-myristate 13-acetate (PMA) | Sigma-Aldrich | p-8139 |
| Puromycin | Invivogen | ant-pr-1 |
| Rotenone | Sigma-Aldrich | R8875-1G |
| RPMI 1640 Medium | Gibco | 21875-034 |
| RPMI 1640 Medium without L-Gln and L-Leu | MP Biomedicals | 091629149 |
| RPMI 1640 Medium without L-Arg, Leu, Lys and without phenol red | Sigma-Aldrich | R1780-500ML |
| Stainless Steel Beads, 7 mm diameter | Qiagen | 69990 |
| Trucount Absolute Counting Tubes IVD | BD Biosciences | 340334 |
| UltraComp eBeads | Invitrogen | 01-2222-41 |
| XF Pro M FluxPak | Agilent Technologies | 103775-100 |
| **Software** | | |
| GraphPad Prism v10.2.3 | GraphPad Software Inc. | www.graphpad.com |
| FlowJo v10.10.0 | BD Life Sciences | www.flowjo.com |
| R language v4.3.2 | R Core Team | |
| RStudio (2023.06.2), | RStudio: Integrated Development for R | www.rstudio.com |
| Bruker IVDr software | Bruker BioSpin | |
| ImageJ v1.54d | Wayne Rasband, NIH | |
| NDP.view2 Image viewing software | Hamamatsu Photonics K.K. | https://www.hamamatsu.com/eu/en/product/life-science-and-medical-systems/digital-slide-scanner/U12388-01.html |
| Adobe Illustrator v28.3 | Adobe Systems | |
| BioRender | | https://BioRender.com |
| **Other** | | |
| 600 MHz IVDr NMR spectrometer | Bruker BioSpin | |
| BD FACSCanto II | BD Biosciences | |
| BD FACS Aria II Cell Sorter | BD Biosciences | |

| Reagent/Resource | Reference or Source | Identifier or Catalog Number |
|---|---|---|
| Mark I 68 A LC Irradiator | JL Shepherd and Associates | |
| Seahorse XF Pro analyzer | Agilent Technologies | |
| Tissue Lyser LT | Qiagen | |
| Zeiss LSM 780 confocal microscopy | Zeiss | |

## Mice

For specific identifiers of mouse lines see Reagents and Tools Table. CD4-Cre, β-actin-eGFP, and Rosa26-floxed-stop-tdTomato (Tmt) mouse lines (all C57BL/6 background, H-2K$^b$, CD45.2) were purchased from The Jackson Laboratory. B6.SJL-$Ptprc^a$ $Pepc^b$/BoyJ (H-2K$^b$, CD45.1) mouse line was purchased from Charles River Laboratories and BALB/cJRj (H-2K$^d$, CD45.2) from Janvier Laboratories. $Slc7a5^{fl/fl}$ mice (C57BL/6 background, H-2K$^b$, CD45.2) were kindly provided by Professor Peter M. Taylor (Dundee University, England) (Poncet et al, 2014). Generation of SLC7A5$^{WT}$ (CD4-Cre$^{+/-}$ $Slc7a5^{fl/wt}$ Tmt$^{fl/wt}$) and SLC7A5$^{\Delta CD4}$ (CD4-Cre$^{+/-}$ $Slc7a5^{fl/fl}$ Tmt$^{fl/wt}$) donor mice were described previously (Cibrian et al, 2020). All experiments were performed using male and female mice of 8–12 weeks kept on a regular 12 h light/dark cycle (7 a.m.–7 p.m. light period), with food and water available ad libitum. All mice were bred and maintained in the pathogen–free animal facilities of the Centro Nacional de Investigaciones Cardiovasculares (CNIC, Madrid, Spain) under the animal care standards of the institution. All experimental procedures with animals were approved by Institutional Animal Care Committee following Spanish and European guidelines (Proex 210, 201.6, 206.1).

## Cell lines and cell culturing

KLN205 cells (lung carcinoma cell line syngenic to BALB/c mice) were obtained from ATCC. KLN205 cells were cultured in 150 mm dishes in MEM medium containing 10% fetal bovine serum (FBS), 2 mM L-Gln, 10 mM HEPES buffer pH 7.2–7.5, 100 U/mL – 100 μg/mL penicillin-streptomycin, and 1:100 non-essential amino acids. Cells were maintained at 37 °C and 5% v/v CO$_2$ in a humidified incubator and were routinely tested for potential mycoplasma contamination. Cells were harvested using trypsin-EDTA and subcultured when they reached confluence. For GVT experiments, the previous day to tumor injection, the KLN205 were passaged to ensure harvest cells in exponential growth the next day.

## Cell preparations for GVHD and GVT experiments

B6.SJL mice or C57BL/6 β-actin-eGFP were used as bone marrow (BM) cell donors, while Tmt$^+$SLC7A5$^{WT}$, Tmt$^+$SLC7A5$^{\Delta CD4}$ or CD45.1$^+$SLC7A5$^{WT}$ mice were used as splenic T cells (STC) donors. BM were depleted of T cells and STC were purified by negative selection using streptavidin magnetic beads (STEMCELL

Technologies) and biotinylated antibodies (see Reagents and Tools Table for clones and catalog references). Cocktail antibodies for STC purification included (dilution 1/100): anti-CD11b, CD11c, CD19, CD45R/B220, CD49b, MHC-class II/I-A/I-E, and TER-119, while BM were T-cell-depleted using anti-CD3 (dilution 1/50). The purity obtained in all the experiments was checked by flow cytometry and exceeded 90% for STC, while T cells were lower than 0.3% in T-cell-depleted BM (TCDBM). For CellTrace Violet (CV, Invitrogen) staining, a final concentration of 3 μM of CV was added to pre-warm (37 °C) STC at $1.5 \times 10^6$ cells/mL in PBS and incubated at 37 °C for 20 min. The excess CV was washed with PBS.

## aGVHD and GVT model

Eight- to twelve-week-old BALB/cJRj recipient mice were γ-ray-irradiated (day 0) with a single dose of 9 Gy from a $^{137}$Cs source at a dose rate of 50 cGy/min (irradiator Mark I 68A LC, JL Shepherd and Associates). BALB/c recipients were intravenously injected with $4 \times 10^6$ allogeneic TCDBM cells from B6.SJL or β-actin-eGFP mice, and $1.5–2 \times 10^6$ allogeneic Tmt$^+$ STC from SLC7A5$^{WT}$ or SLC7A5$^{\Delta CD4}$ mice, obtained as described above. For the competitive migration experiment, BALB/c recipient mice were intravenously injected with $4 \times 10^6$ allogeneic TCDBM cells from β-actin-eGFP mice, $2 \times 10^6$ allogeneic Tmt$^+$ STC from SLC7A5$^{\Delta CD4}$ mice and $2 \times 10^6$ allogeneic CD45.1$^+$ STC SLC7A5$^{WT}$ cells from B6.SJL mice, obtained as described above. For survival experiments, BALB/c mice were monitored weekly for physical appearance and body weight. Survival curves were assessed daily for at least 80 days. None mouse was excluded from survival analyses. For flow cytometry and histological analysis, mice were euthanized on days 3, 7, or 40 after the transplant. For cell proliferation assays, STC were stained as above described with CV previously to be infused into the host BALB/c mice ($2 \times 10^6$ CV-labeled STC in SLC7A5$^{WT}$ and SLC7A5$^{\Delta CD4}$ comparison and $5 \times 10^6$ for Leu-deprived diet experiment).

L-Leu-deprived or (-)Leu diet (A05080202i) and isocaloric regular diet (A10021B) were obtained from Research Diets, Inc. BALB/c recipient mice were acclimated to regular or (-)Leu diets for at least two weeks before irradiation and transplantation. Once the diets were changed, they were maintained uninterruptedly until the end point of the experiments. However, (-)Leu diet-fed mice were given access to one pellet per mice of regular diet every 15 days during survival experiments, to avoid possible adverse effects of severe nutritional deficiency. After 2–3 weeks of formulated diets, all BALB/c mice were irradiated and injected with $4 \times 10^6$ allogeneic CD45.1$^+$ TCDBM cells and $1.5–2 \times 10^6$ ($5 \times 10^6$ in proliferation assay) allogeneic Tmt$^+$ SLC7A5$^{WT}$ STC cells.

For GVT model, $10^6$ KLN205 cells in 100 uL of PBS were subcutaneously (sc.) inoculated into the left flank of the BALB/cJRj recipient mice, the following day of transplantation. BALB/c irradiated and transplanted with complete BM from BALB/c donors fed with regular or (-)Leu diet were used as controls. Tumor sizes were measured with an electronic caliper every 2–5 days by a specialized researcher blinded to experimental groups. Dead mice were excluded from the tumor growth analysis. Tumor volumes were calculated using the formula volume = width$^2$ × length/2 and presented as mean ± standard deviation (SD) mm$^3$ (Bartolome-Izquierdo et al, 2017). Twenty-five days later, spleen and mesenteric lymph nodes (mLN) were retrieved from animals

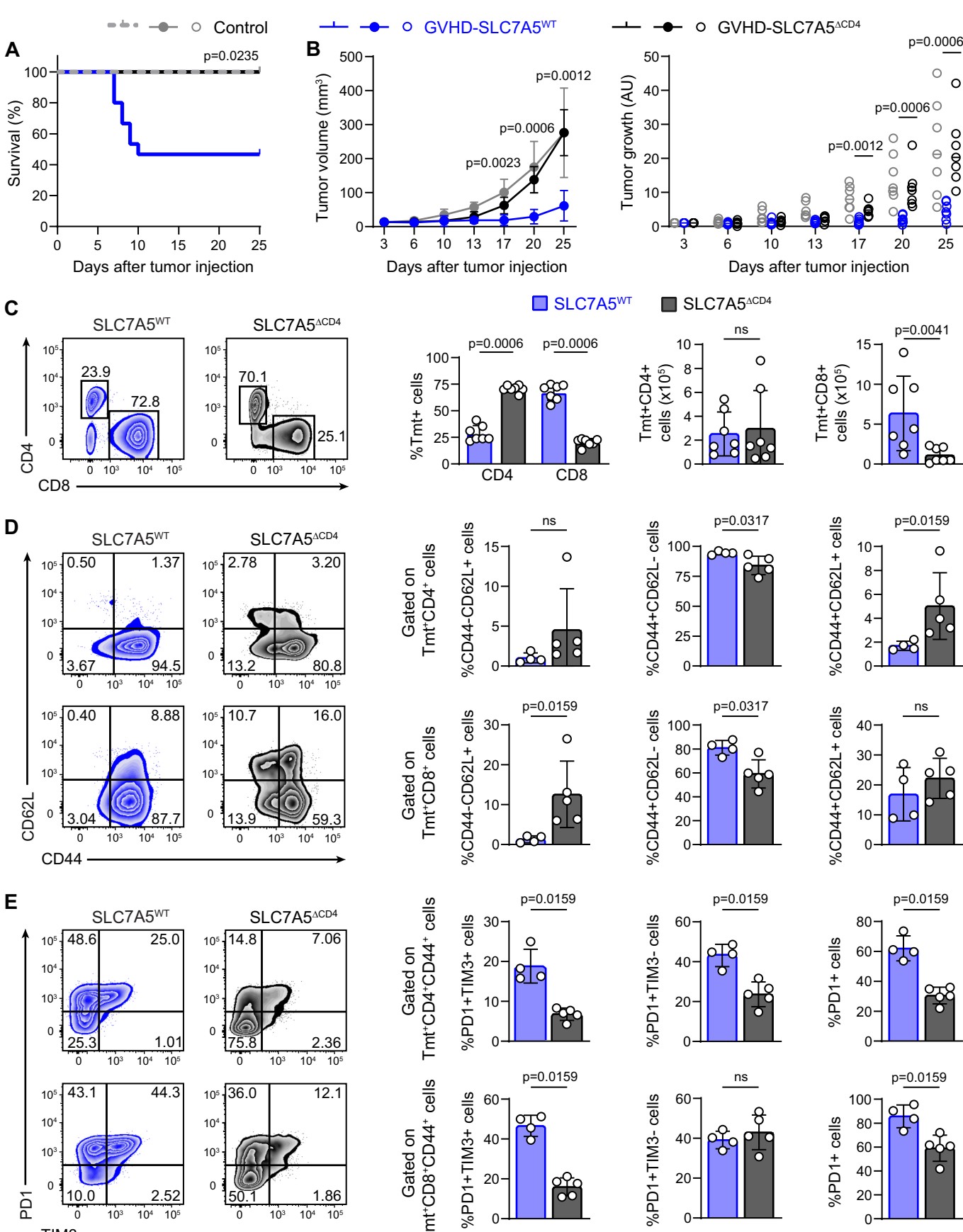

for flow cytometry analyses and tumors were processed for histology.

## Mitochondrial respiration and glycolysis assays

The oxygen consumption rate (OCR) and the extracellular acidification rate (ECAR) were measured using an XF Pro extracellular flux analyzer (Seahorse Bioscience; XFPro M FluxPak Agilent Technologies). CD4+ T cells purified from spleen and lymph node suspension of SLC7A5WT and SLC7A5ΔCD4 mice were seeded at 10^6 cells/mL on a 24-well plate coated with 5 μg/mL anti-CD3 (145-2C11, Tonbo) and 1 μg/mL anti-CD28 (37.51, Tonbo), in complete RPMI 1640 medium (supplemented with 5% heat-inactivated FBS, 2 mM L-Gln, 10 mM HEPES buffer pH 7.2–7.5, 100 U/mL – 100 μg/mL penicillin-streptomycin, 1 mM Na-pyruvate and 50 μM 2-mercaptoethanol) or completed RPMI 1640 medium without L-Gln and L-Leu, supplemented with 2 mM L-Gln. Rapamycin (100 nM) was added during the last 24 h prior to metabolic measurement, in both mediums. After 48 h of activation, cells were collected and seeded ($2 \times 10^5$ cells per well) in poly-l-Lys (50 ng/ml) pre-coated Seahorse culture plates, using red-phenol-free complete RPMI medium supplemented with 1 mM Na-pyruvate, 1 mM L-Gln and 25 mM glucose, or red-phenol free RPMI medium without Leu, Arg and Lys, supplemented with 1 mM Na-pyruvate, 1 mM L-Gln, 25 mM glucose, 1 mM L-Arg and 0.2 mM L-Lys. The mitochondrial stress assay comprised the sequential injection of oligomycin (13.5 μM), BAM15 (25 μM), and rotenone (5.5 μM), plus antimycin A (5.5 μM). The glycolysis stress assay involved sequential injections of glucose (100 mM), oligomycin (20 μM), and 2-deoxy-D-glucose (2-DG; 500 mM). Three consecutive mix and measure steps were performed for resting conditions and after each injection (3 min each). At least four independent donors and three technical replicates were included by assay.

## Flow cytometry

Spleen and mLN were grated through a 70 μm pores cell strainer (Falcon) to obtain single-cell suspensions. Erythrocytes were lysed with 1 mL ACK buffer lysis (Lonza) for 3 min at RT only in spleen suspensions. Colons were dissected longitudinally, washed several times with cold 5% FBS-RPMI and cut into small pieces. Colon pieces were then digested in 5% FBS-RPMI containing 250 μg/ml liberase TM (Roche) and 100 μg/ml DNAse I (Roche) for 30 min at

37 °C under constant stirring. Tissue digestion was stopped by the addition of 10% FBS-RPMI. The digested suspension was filtered through a 70 μm cell strainer and the remaining undigested tissue was manually mashed through the strainer and further purified by a 40–80% gradient of Percoll. The ears were separated into inner and outer halves and cut into small pieces with scissors. Ears were digested in 2% FBS-RPMI containing 125 μg/ml liberase TM, 100 μg/ml DNAse I, and 500 μg/ml collagenase IV for 30 min at 37 °C under constant stirring. The digestion reaction was quenched by adding PBS containing 5% FBS and 5 mM EDTA (PFE buffer), and the digested suspension was filtered through a 70-μm cell strainer. The remaining undigested tissue was mechanically disrupted using 7 mm Stainless Steel Beads in a Tissue Lyser LT (Qiagen) with a 3-min cycle at 20 Hz at RT and filtered again through the same strainer pooling the complete cell suspension of the tissue. Lungs were digested in 2% FBS-RPMI containing 0.25 mg/ml Liberase TL and 50 mg/ml DNAse I for 20 min, at 37 °C, with shaking. Tissue digestion was stopped by the addition of PFE buffer, and the digested suspension was filtered through a 70 μm cell strainer.

For cytokine staining of in vivo GVHD, spleen, and mLN were retrieved from SLC7A5WT → BALB/c and SLC7A5ΔCD4 → BALB/c mice. Cell suspensions were obtained as explained above and seeded at $1.5 \times 10^6$ cells/mL in complete RPMI medium (RPMI supplemented with 5% heat-inactivated FBS, 2 mM L-Gln, 10 mM HEPES buffer pH 7.2–7.5, 100 U/mL – 100 μg/mL penicillin-streptomycin, 1 mM Na-pyruvate and 50 μM BME) on a 96-U-well plate coated with 2 μg/mL anti-CD3 (145-2C11, Tonbo) and 0.5 μg/mL anti-CD28 (37.51, Tonbo). Cells were incubated for 24 h at 37 °C and 5% $CO_2$ and were further stimulated in the same wells with 50 ng/mL PMA (Sigma), 500 ng/mL ionomycin (Sigma), and 1:1000 GolgiStop (monensin, BD) for 4 h.

For in vitro cytokine staining, spleen and mLN were retrieved from SLC7A5WT and SLC7A5ΔCD4 mice. Cell suspensions were obtained as explained above and incubated ($2 \times 10^6$ cells/ml) with 2 μg/mL concanavalin A in complete or Leu-deprived RPMI medium (supplemented as indicated before) on 6-well plates for 48 h at 37 °C and 5% $CO_2$. After discarding dead cells using Ficoll separation solution gradient, cells were further stimulated with 50 ng/mL PMA (Sigma), 500 ng/mL ionomycin (Sigma), and 5 μg/ml of brefeldin for 4 h.

The staining panels always included Fc γ-receptor-blocking antibody (αCD16/αCD32, Tonbo), Trucount Absolute Counting beads (BD) to quantify the total number of cells, and LIVE/DEAD

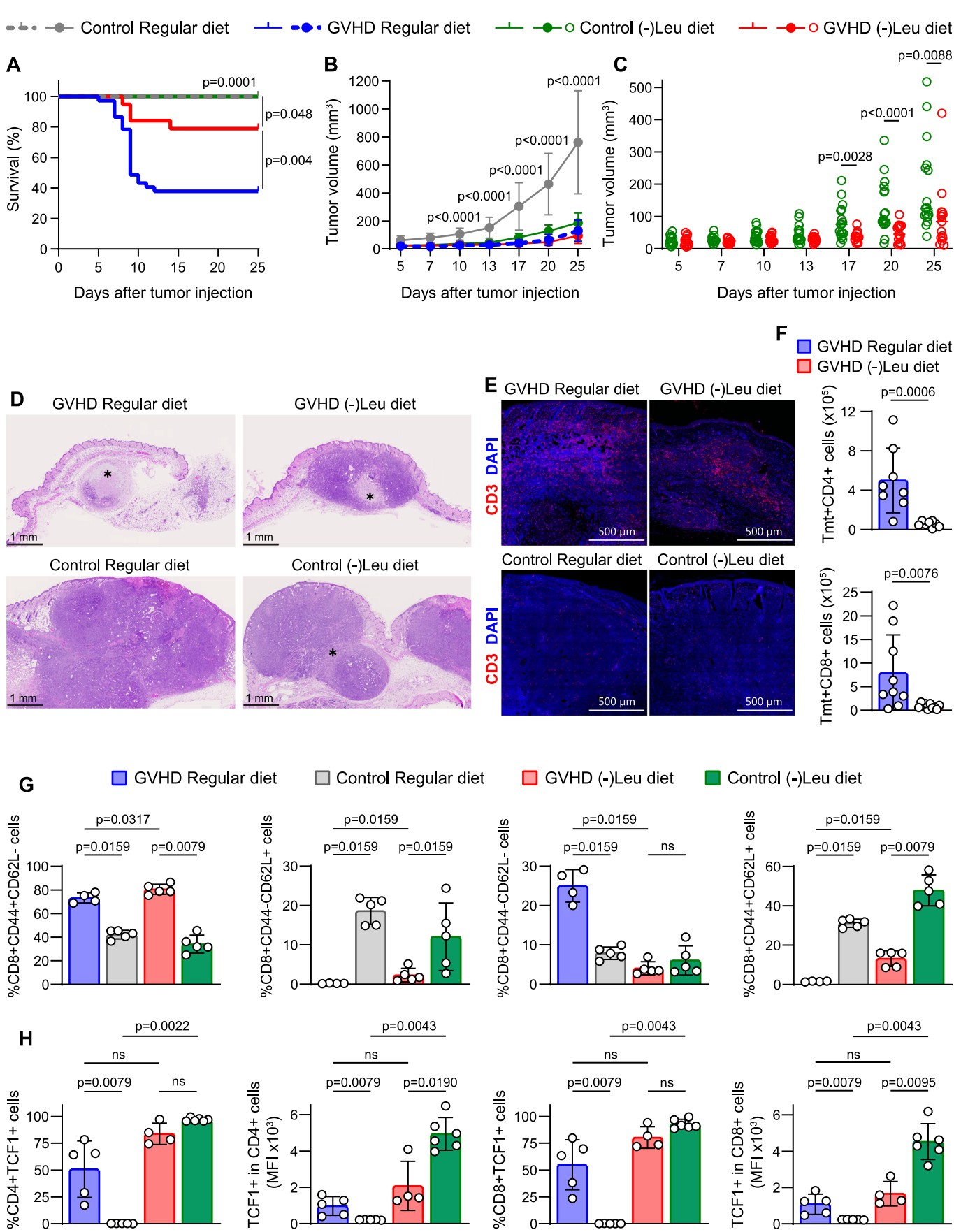

**Figure 8. Dietary restriction of Leu maintains GVT response.**

(A) Survival curve of BALB/c recipients mice transplanted with BM from BALB/c mice fed with regular (control regular diet, gray, $N = 16$) or (-)Leu diet (control (-)Leu diet, green, $N = 17$) and BALB/c transplanted with TCDBM from CD45.1$^+$ donor mice and STC from SLC7A5$^{WT}$ fed with regular (GVHD regular diet, blue, $N = 37$) or (-)Leu diet (GVHD (-)Leu diet, red, $N = 19$) subcutaneously (sc.) inoculated with $10^6$ KLN205 cells. A pool of three independent experiments is shown. Long-rank (Mantel-Cox) and Cehan-Breslow-Wilcoxon test. Survival was evaluated from the first day of tumor injection until death or sacrifice. (B, C) Mean of volume (B) of subcutaneous tumors measured with an electronic caliper of all groups and individual tumor volume (C) of (-)Leu diet-fed groups. A pool of three independent experiments is shown. Only live animals at end point are represented: control regular diet $N = 16$, control (-)Leu diet $N = 17$, GVHD regular diet $N = 14$, GVHD (-)Leu $N = 15$. $P$ values are calculated from GVHD regular diet versus control regular diet on (B), and from GVHD (-)Leu diet versus control (-)Leu diet on (C), two-tailed Mann–Whitney's test. (D) Representative H&E-sections of the tumors after 25 d of sc. tumor injection. Scale bars $= 1$ mm. Asterisks (*) indicate focal necrotic areas. (E) Representative immunofluorescence of CD3 (red) in tumor sections at 25 d after tumor injection. Nuclei were stained with DAPI (blue). Scale bars $= 500$ μm. (F) Absolute numbers of Tmt$^+$CD4$^+$ and Tmt$^+$CD8$^+$ T cells in spleen of SLC7A5$^{WT}$ → BALB/c mice fed with regular or (-)Leu diet after 25 d of sc. tumor injection. (G) Frequencies of effector memory (CD44$^+$CD62L$^-$), naive (CD44$^-$CD62L$^+$), double negative (CD44$^-$CD62L$^-$) and central memory (CD44$^+$CD62L$^+$) T cells gated on spleen Tmt$^+$CD8$^+$ T cells in GVHD groups and CD8$^+$ cells in control groups, after 25 d of sc. tumor injection. (H) Frequencies and mean fluorescence intensity (MFI) of TCF-1 of Tmt$^+$CD4$^+$CD44$^+$ or Tmt$^+$CD8$^+$CD44$^+$ cells from spleen of GVHD groups and CD4$^+$CD44$^+$ or CD8$^+$CD44$^+$ cells of spleen from control groups after 25 d of sc. tumor injection. (F–H): $N = 4$–7 per group, mean ± SD, two-tailed Mann–Whitney's test, ns: not significant. Source data are available online for this figure.

Fixable Yellow Dead Cell Stain to label dead cells (Invitrogen), except for apoptosis and proliferation staining. Single-cell suspensions were subsequently stained with 1:200 dilutions of the respective surface marker antibodies (see Reagents and Tools Table for clones and catalog references) for 30 min at 4 °C prepared in PFE buffer: CD3ε, CD4, CD8α, CD11b, CD44, CD45.1, CD45.2, CD62L, CD69, Ly6C, Ly6G, TCRγδ, PD-1, and TIM3. When intracellular antigens were targeted, cells were pre-fixed with 1% paraformaldehyde in PBS for 30 min at 4 °C. Cells were subsequently fixed, washed, and incubated with the solutions provided by the BD Cytofix/cytoperm fixation/permeabilization kit for intracellular staining of cytokines (IFNγ, granzyme B) or by the FoxP3/Transcription Factor Staining Buffer Set (eBioscience) for intranuclear antigens (Tbet, TCF1, and FoxP3). In both cases, cells were fixed with the corresponding fixation/permeabilization buffer for 30 min at 4 °C and stained for 1–2 h or overnight (ON) at 4 °C with 1:100 dilutions of the intracellular antibodies prepared in the corresponding permeabilization buffer: FoxP3, granzyme B, Tbet, IFNγ, and TCF1. For apoptosis assessment, mLN suspensions obtained at 7 days post-transplant were stained with Annexin V-CF Blue 7-AAD Apoptosis Staining/Detection Kit (Abcam) according to the manufacturer's protocol.

After staining, cells were washed and acquired in a FACS Canto 3L (BD Biosciences). The software used for the acquisition was BD FACSDiva (BD Biosciences) and samples were analyzed with FlowJo v10.10.0 (BD Life Sciences). UltraComp eBeads (Invitrogen) were used for compensation.

## Protein synthesis assay

CD8$^+$ T cells were purified by cell sorting (BD FACS Aria II Cell Sorter) from spleen and mLN suspensions of SLC7A5$^{WT}$ and SLC7A5$^{ΔCD4}$ mice and seeded at $10^6$ cells/mL in complete or Leu-deprived RPMI medium (supplemented as indicated before) on a 96-U-well plate coated with 5 μg/mL anti-CD3 (145-2C11, Tonbo) and 2 μg/mL anti-CD28 (37.51, Tonbo). Cells were incubated 24 h at 37 °C and 5% $CO_2$ and were further stimulated with 50 ng/mL PMA (Sigma) and 500 ng/mL ionomycin (Sigma) for 3 h. To inhibit protein synthesis, puromycin and cycloheximide (10 μg/ml each one) were added 1 h before the stimulation while the O-propargyl-puromycin (OPP) (20 μM) was added in the last 30 min of stimulation. The click reaction to detect OPP

was performed according to the manufacturer's instructions (Life Technologies).

## Histology

Histological samples were obtained from the spleen and colon on days 7 and 40 post-transplant. Tumors were collected after 25 days of transplant. Tissue samples were fixed in 10% Formalin for 72 h and then processed for paraffin embedment. Tissue blocks were ground in 4-μm-sections and stained with hematoxylin and eosin (H&E). Microscope slides were digitalized and analyzed with NDP.view2 software.

For immunofluorescence, spleen, colon, and tumor slides were deparaffinized and boiled in antigen retrieval solution (10 mM sodium citrate, 0.05% Tween 20, pH 6.0). Tissue sections were blocked in PBS containing 10% donkey serum and 2% BSA and then incubated with primary antibodies ON at 4 °C followed by fluorophore-conjugated secondary antibodies (dilution 1/500) (see Reagents and Tools Table for catalog references). Primary antibodies used (dilution 1/200) were: CD3ε, CD8α, GFP and Tmt. Nuclei were counterstained with DAPI. Images were obtained with Zeiss LSM 780 confocal microscopy with 40x OIL objective and processed with ImageJ.

## Sample preparation and acquisition NMR metabolomics

Serum samples were handled under the same standard operating procedures and stored at −80 °C until analysis. They were prepared manually, 30 μL of serum and 70 μL of Mili-Q were mixed with serum buffer (75 mM $Na_2HPO_4$, 2 mM $NaN_3$, 4.6 mM sodium trimethylsilyl propionate-[2,2,3,3-$^2H_4$] (TSP) in 10% $D_2O$, pH 7.4 ± 0.1) in a 1:1 (v/v) ratio for a final volume of 200 μL into the 3 mm NMR tube. NMR measurements were done in a 600 MHz IVDr (Bruker BioSpin, Silberstreifen, Germany) with a tempered SampleJet automatic sample changer mounted on it and a double resonance broadband probe (BBI) probe head with a z gradient coil and BOSS-III shim system. NMR sample tubes were stored inside the SampleJet at 5 °C until measurement. Every morning the spectrometer was calibrated with three different samples: methanol, QuantRef and Sucrose to check the temperature (310 K), the quantification performance and optimal shimming, respectively, following strict standard operation procedures, as previously

described (Gil-Redondo et al, 2022). Three different ¹H NMR experiments were recorded in all samples: a standard one-dimensional (1D) ¹H NOESY spectrum (noesygppr1d) with water presaturation, a 1D ¹H Carr–Purcell–Meiboom–Gill (CPMG) experiment (cpmgpr1d) implementing a T2 filter to suppress the broad signals of proteins and other macromolecules, and a J-Edited DiFFusional Pulsed Gradient Echo Experiment (JEDI-PGPE) (Nitschke et al, 2022).

Quantification of serum metabolites and inflammation parameters from ¹H-NMR spectra was performed using Bruker IVDr software, specifically B.I.Quant-PS 2.0.0 for the quantification of 39 serum metabolites in mmol/L units, and PhenoRisk PACS RuO 1.0.0 for the quantification of five inflammation parameters in procedure defined units (p.d.u.). The complete list of quantified variables is available in Table EV1.

## Statistical analysis

Data were analyzed with GraphPad Prism v10 software (GraphPad Software Inc.). Graphs show the individual distribution of each sample, and all the statistical variables were expressed as mean ± standard deviation (SD). Mice were always randomly assigned to experimental groups and sample sizes were estimated according previous studies (Cibrian et al, 2020). U-Mann–Whitney's test for two group comparisons or Kruskal–Wallis with Dunn's post hoc test for multiple comparisons test were used. Long-rank (Mantel-Cox) test and Cehan-Breslow-Wilcoxon test were used for the analysis of the Kaplan–Meier curve (survival curve). Differences were considered statistically significant at $p \leq 0.05$ and were indicated as exact $p$-values in figures. Non-significant comparisons ($p > 0.05$) were designed as 'ns' and were indicated in the graph exclusively when they were relevant, otherwise were omitted. All statistical tests used and $n$ numbers have been mentioned in Figures legends. Graphs were generated using GraphPad Prism v10.2.3 and assembled with Adobe Illustrator v28.3 (Adobe Systems).

To find out the impact of irradiation, transplant, and the activation of allogenic T cells on the metabolomic profile for each of the diets (regular and L-Leu-free diet), a univariate analysis was conducted against the regular diet group (BALB/c mice non-irradiated and non-transplanted). For each metabolite within each diet or transplant, the effect size was determined by calculating the fold-change (FC, mean of the group being analyzed divided by the mean of the control group). This is represented using its base 2 logarithm ($\log_2$FC) for ease of interpretation. Statistical significance is derived from the unadjusted $P$-value obtained through a student's t-test. $P$-values below 0.05 are considered statistically significant. A forest plot illustrates the effect sizes, along with their standard errors and statistical significance, for each variable within each diet type and each type of allogenic T cells, always in reference to the regular diet group. Standard error was computed through bootstrapping simulations. Unsupervised multivariate analysis via Principal Component Analysis (PCA) reveals the positioning of each group by plotting their first two principal components on a scores plot. To better observe the metabolomic pattern of each individual and their groupings, auto-scaled variables (subtracting the mean and dividing by the standard deviation) are projected onto a heatmap. This heatmap organizes both rows (individuals) and columns (variables) through hierarchical clustering using Euclidean distance and Ward D2 as the linkage method. All

analyses were conducted using R language (4.3.2) with RStudio (2023.06.2), employing the following R packages: tidyverse (2.0.0), ggforce (0.4.1), factoextra (1.0.7), ComplexHeatmap (2.16.0), and ggforestplot (0.1.0).

### The paper explained

#### Problem

Acute graft-versus-host disease (aGVHD) is a common life-threatening complication after allogeneic hematopoietic cell transplantation. Administration of immunosuppressive regimes is the main therapy for aGVHD, but entails detrimental complications for patients and the loss of the graft versus-tumor (GVT) effect, increasing the risk of tumor relapse. Therefore, strategies that successfully control allogenic T cell-mediated inflammation and prevent tumor growth are required. The L-Leu amino acid transporter SLC7A5 has become an important target in inflammation and cancer, but its role in aGVHD and GVT has not been explored before.

#### Results

Our research demonstrates that genetic deletion of SLC7A5 as well as dietary restriction of L-Leu is sufficient to control aGVHD by controlling allogenic T cell expansion, survival, and cytokine release. However, genetic deletion of SLC7A5 in activated T cells severely compromises GVT response, while dietary restriction of L-Leu controls tumor growth and further maintains GVT response. Mechanistically, L-Leu deprivation reduced mTORC1 and glycolysis capacity of activated T cells but does not affect OXPHOS and ATP content, as observed with genetic deletion of SLC7A5. These findings reveal SLC7A5-dependent changes in activated T cells which are not caused by L-Leu deprivation. Moreover, our work demonstrates that both strategies, genetic deletion of SLC7A5 in allogenic T cells and dietary restriction of L-Leu, prevent the systemic increase of ketone body acetoacetate and beta-hydroxybutyric acid, which could be regulating aGVHD and GVT response.

#### Impact

These findings suggest that restriction of dietary L-Leu intake is a promising therapeutic strategy to control SLC7A5-mediated allogenic T cell expansion and aGVHD, reducing tumor growth and maintaining GVT response. In addition, our data provide evidence that L-Leu is the preferential substrate oxidized to ketone bodies acetoacetate and β-hydroxybutyric acid during aGVHD, suggesting the relevance of these metabolites as biomarkers of allogenic T cell expansion and predictors of transplant rejection.

## Data availability

Primary dataset from metabolomics studies, corresponding to Fig. 5, is available here: https://data.mendeley.com/datasets/9h2j5zwjst/1; https://doi.org/10.17632/9h2j5zwjst.1. Source data for microscopy images of Figs. 1, 2 and 8 are available in BioImage with the accession number S-BIAD1834 or at https://www.ebi.ac.uk/biostudies/bioimages/studies/S-BIAD1834. Flow cytometry (.fcs files) for Figs. 3 and 7 are provided in BioStudies with the accession number S-BSST1964 or at https://www.ebi.ac.uk/biostudies/studies/S-BSST1964.

The source data of this paper are collected in the following database record: biostudies:S-SCDT-10_1038-S44321-025-00250-2.

# Peer review information

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

## Acknowledgements

This study has been funded to DC by Instituto de Salud Carlos III (ISCIII) through the project "PI22/01842" and co-funded by the European Union (FEDER); to FS-M by Ministerio de Ciencia e Innovación (grant nº. PID2023-149541OB-I00), by AECC grant (PRYCO223002PEIN), CIBER Cardiovascular from the Instituto de Salud Carlos III (Fondo de Investigación Sanitaria del Instituto de Salud Carlos III with co-funding from FEDER); to FS-M and NBM-C by Comunidad de Madrid (grant nº S2022/BMD-7209-INTEGRAMUNE-CM) and La Caixa Banking Foundation (LCF/PR/HR23/52430018); to JA by Ministerio de Ciencia e Innovación (grant nº PID2019-106371RB-I00) and CIBER Cardiovascular from the Instituto de Salud Carlos III (Fondo de Investigación Sanitaria del Instituto de Salud Carlos III with co-funding from FEDER). NF-G was supported by the Formación de Profesorado Universitario (FPU) program (FPU16/03953) from Ministerio de Universidades and INTEGRAMUNE; NF-G and SL-A by Programa Investigo (09-PIN1-00015.6/2022) from Comunidad de Madrid funded by European Union NextGenerationEU/Plan de Recuperación, Transformación y Resiliencia (PRTR) de España. BA was supported by Programa Investigo (2022-C23.I01.P03.S0020-0000031) from Ministerio de Trabajo y Economía Social, Servicio Público de Empleo Estatal (SEPE) funded by European Union NextGenerationEU/ PRTR de España. RC-G was supported by Ayudas para contratos Juan de la Cierva-formación 2021 (FJC2021-047282-I) from Ministerio de Ciencia e Innovación. AR-G was supported by Formación de Personal Investigador (FPI) program (FPI-SAF2017-82886-R) from Ministerio de Ciencia e Innovación. DC is supported by Miguel Servet Program (CP21/00135), funded by the Instituto de Salud Carlos III (ISCIII) and co-funded by the European Union (FSE+). The CNIC is supported by the Instituto de Salud Carlos III (ISCIII), the Ministerio de Ciencia e Innovación (MCIN), the Pro CNIC Foundation, and Severo Ochoa Program (grant CEX2020-001041-S funded by MICIN/AEI/10.13039/501100011033). The microscopy experiments were performed at the Microscopy and Dynamic Imaging Unit, CNIC, ICTS-ReDib, co-funded by MCIN/AEI/10.13039/501100011033. Funding agencies did not intervene in the design of the studies, with no copyright over the study. We thank the CNIC facilities, especially the cytometry, microscopy, and animal care-related personnel.

## Author contributions

**Nieves Fernández-Gallego**: Conceptualization; Formal analysis; Investigation; Visualization; Methodology; Writing—original draft; Writing—review and editing. **Blanca Anega**: Conceptualization; Formal analysis; Investigation; Visualization; Methodology; Writing—original draft; Writing—review and editing. **Susana Luengo-Arias**: Investigation; Visualization; Writing—review and editing. **Maider Bizkarguenaga**: Investigation; Visualization; Methodology; Writing—review and editing. **Rubén Gil-Redondo**: Data curation; Formal analysis; Investigation; Visualization; Methodology; Writing—review and editing. **Nieves Embade**: Investigation; Visualization; Methodology; Writing—review and editing. **Laura Navarrete-Arias**: Investigation; Visualization; Writing—review and editing. **Marta Ramírez-Huesca**: Investigation; Visualization; Writing—review and editing. **Emigdio Álvarez-Corrales**: Investigation; Visualization; Writing—review and editing. **Sara G Dosil**: Investigation; Visualization; Writing—review and editing. **Raquel Castillo-González**: Investigation; Visualization; Writing—review and editing. **Amelia Rojas-Gomez**: Investigation; Visualization; Writing—review and editing. **Inés Espeleta**: Investigation; Visualization; Writing—review and editing. **Sara Martínez-Martínez**: Resources; Investigation; Methodology; Writing—review and editing. **Arantzazu Alfranca**: Resources; Methodology; Writing—review and editing. **Virginia G de Yebenes**: Resources; Methodology; Writing—review and editing. **Noa Beatriz Martín-Cófreces**: Resources; Writing—review and editing. **Julián Aragonés**: Resources; Writing—review and editing. **Pilar Martin**: Resources; Writing—review and editing. **Oscar Millet**: Conceptualization; Supervision; Methodology; Writing—review and editing. **Francisco Sánchez-Madrid**: Resources; Funding acquisition; Writing—original draft; Writing—review and editing. **Danay Cibrian**: Conceptualization; Formal analysis; Supervision; Funding acquisition; Investigation; Visualization; Methodology; Writing—original draft; Project administration; Writing—review and editing.

Source data underlying figure panels in this paper may have individual authorship assigned. Where available, figure panel/source data authorship is listed in the following database record: biostudies:S-SCDT-10_1038-S44321-025-00250-2.

## Disclosure and competing interests statement

The authors declare no competing interests.

# Expanded View Figures

**Figure EV1.  Effect of SLC7A5 genetic deletion in T cells in aGVHD model.**

(A) Workflow for aGVHD model. STC: splenic T cell; TCDBM: T-cell-depleted bone marrow; Tmt: Tomato. (B) Representative images of the macroscopic appearance of SLC7A5$^{WT}$ → BALB/c and SLC7A5$^{\Delta CD4}$ → BALB/c mice at 40 d after transplant. (C) Representative density plots of Ly6C$^+$ (monocytes) and Ly6C$^+$Ly6G$^+$ (neutrophils) populations gated on CD11b$^+$ (myeloid) in colon of SLC7A5$^{WT}$ → BALB/c and SLC7A5$^{\Delta CD4}$ → BALB/c mice at 7 d post-transplant. (D) Representative density plots of total CD4$^+$ and CD8$^+$ populations gated on CD3$^+$ cells (left) and density plots of GFP vs Tmt gated on CD4$^+$ (middle) and CD8$^+$ (right) T cells in mesenteric lymph nodes (mLN) of SLC7A5$^{WT}$ → BALB/c and SLC7A5$^{\Delta CD4}$ → BALB/c mice at 7 d post-transplant. (E) Total count (left), GFP$^+$ (donor BM origin, middle) and GFP$^-$Tmt$^-$ (host origin, right) of CD4$^+$ (top) and CD8$^+$ (bottom) cells in mLN and spleen of SLC7A5$^{WT}$ → BALB/c and SLC7A5$^{\Delta CD4}$ → BALB/c mice at 7 d post-transplant. A representative experiment of three is shown. $N = 4$–5 per group, mean ± SD, two-tailed Mann–Whitney's test, ns: not significant.

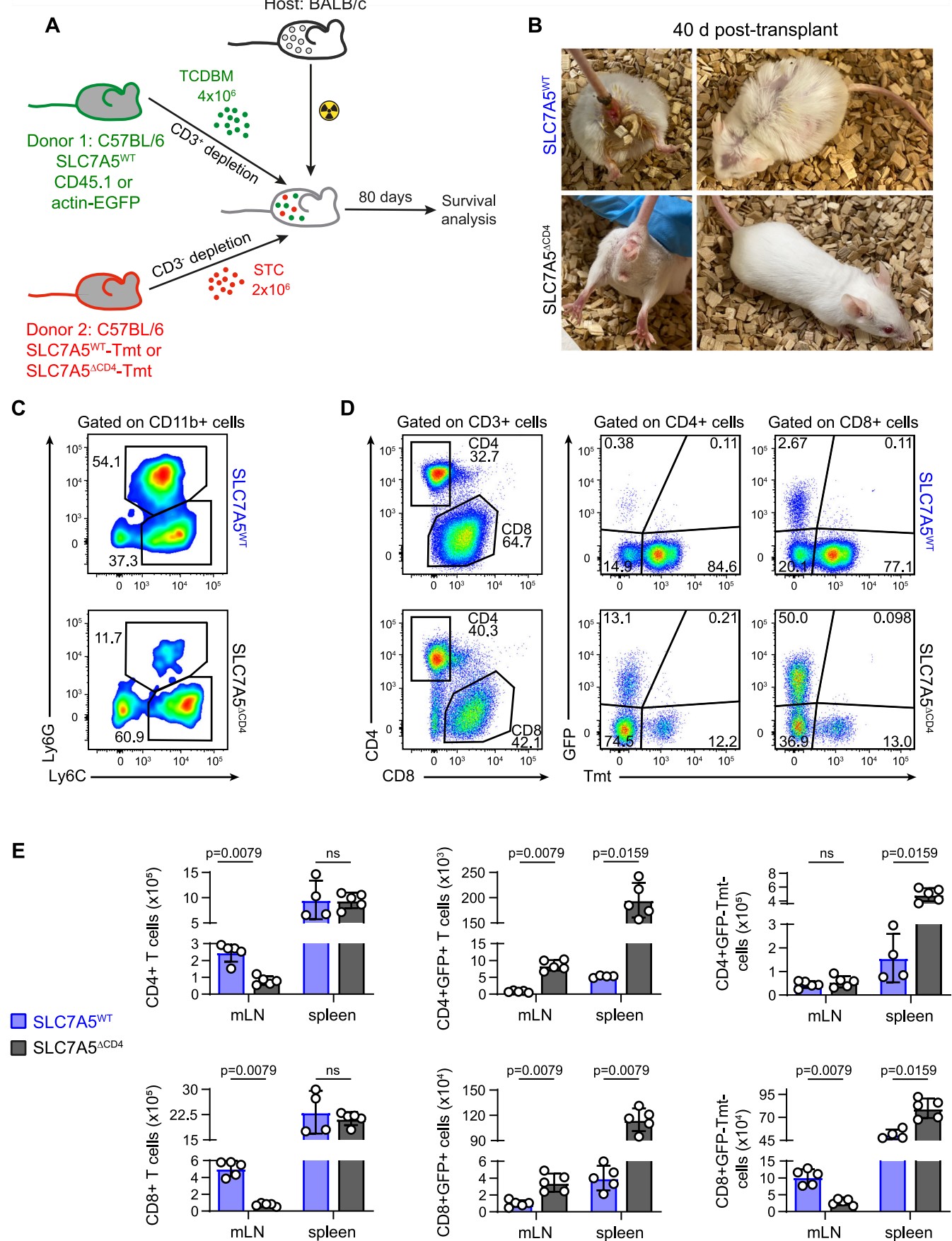

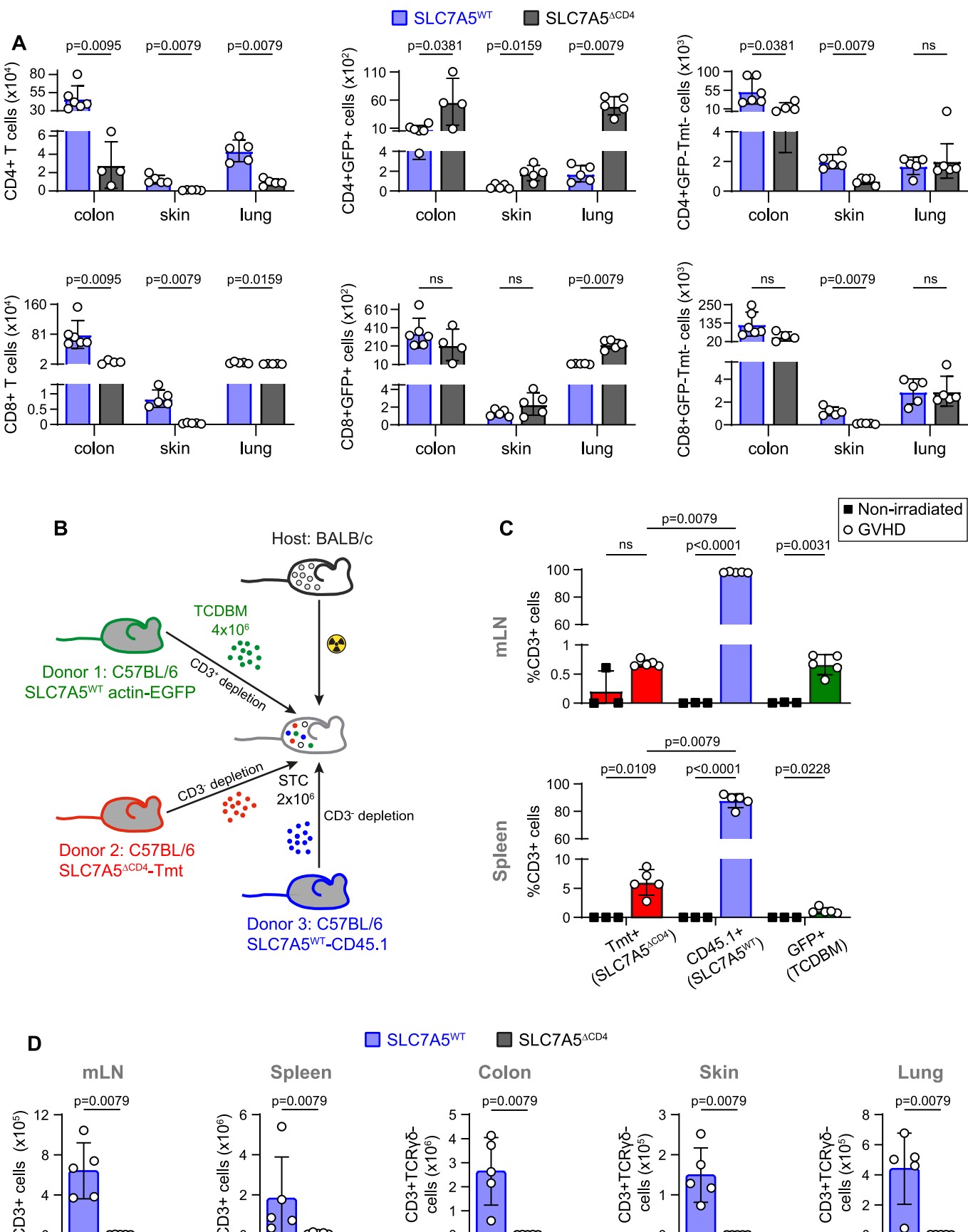

**Figure EV2.  Role of SLC7A5 in T cell expansion and migration in aGVHD model.**

(A) Total count (left), GFP$^+$ (donor BM origin, middle) and GFP$^-$Tmt$^-$ (host origin, right) of CD4$^+$ (top) and CD8$^+$ (bottom) cells in colon, skin and lung of SLC7A5$^{WT}$ → BALB/c and SLC7A5$^{\Delta CD4}$ → BALB/c mice at 7 d post-transplant. (B) Workflow for competitive migration assay. STC: splenic T cell; TCDBM: T-cell-depleted bone marrow; Tmt: Tomato. (C) Percentages of CD3$^+$ cells expressing Tmt$^+$ (STC-derived from SLC7A5$^{\Delta CD4}$ mice donor, red), CD45.1$^+$ (STC-derived from SLC7A5$^{WT}$ B6.SJL mice donor, blue), or GFP$^+$ (TCDBM-derived from β-actin-eGFP mice donor, green) in mLN and spleen from non-irradiated (control) and irradiated (GVHD) BALB/c recipient mice in competitive migration assay. $N = 3$–5 per group, mean ± SD, Kruskal–Wallis test with Dunn's post-test for control and GVHD comparison or two-tailed Mann–Whitney's test for Tmt$^+$ and CD45.1$^+$ of GVHD group comparison. (D) Absolute numbers of CD3 T cells SLC7A5$^{WT}$-CD45.1$^+$ and SLC7A5$^{\Delta CD4}$-Tmt$^+$ in different tissues of GVHD mice of the competitive migration assay. A, D: $N = 4$–6 per group, mean ± SD, two-tailed Mann–Whitney's test. STC splenic T cell, TCDBM T-cell-depleted bone marrow, Tmt Tomato.

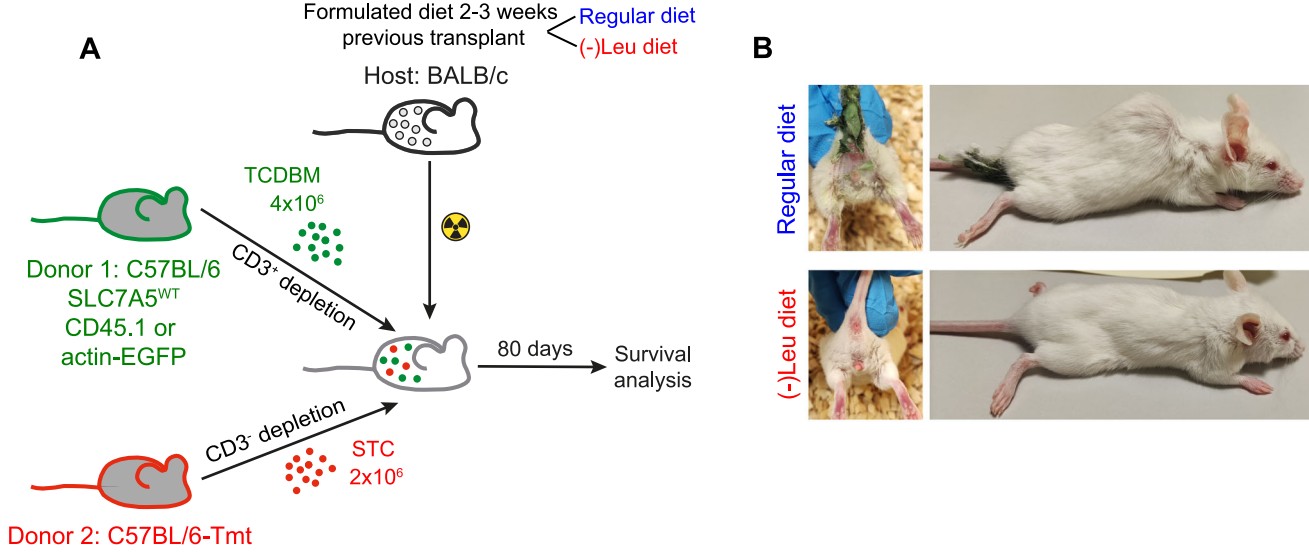

**A**

Formulated diet 2-3 weeks previous transplant — Regular diet / (-)Leu diet

Host: BALB/c

Donor 1: C57BL/6 SLC7A5^WT CD45.1 or actin-EGFP

TCDBM 4×10^6
CD3^+ depletion

80 days → Survival analysis

CD3^- depletion
STC 2×10^6

Donor 2: C57BL/6-Tmt only **SLC7A5^WT**

**B**

Regular diet

(-)Leu diet

**C**   Gated on CD4^+ cells   Gated on CD8^+ cells

SLC7A5^WT

CD4^+: 3.07 | 2.95 | 0.63 | 93.3
CD8^+: 1.00 | 1.23 | 0.32 | 97.4

SLC7A5^ΔCD4

CD4^+: 39.3 | 15.5 | 2.77 | 42.4
CD8^+: 24.6 | 11.4 | 2.28 | 61.7

SLC7A5^WT (-)Leu diet

CD4^+: 2.49 | 3.65 | 0.44 | 93.4
CD8^+: 0.50 | 3.85 | 0.68 | 95.0

CD62L / CD44

**D**   Gated on CD4^+ cells   Gated on CD8^+ cells

SLC7A5^WT
CD4: 16.4 | CD8: 11.6

SLC7A5^ΔCD4
CD4: 23.1 | CD8: 32.8

SLC7A5^WT (-)Leu diet
CD4: 30.4 | CD8: 25.0

CD69 / CD4 — CD8

**E**   Gated on Tmt^+ cells

SLC7A5^WT
0.024 | 0.34 | 69.5 | 30.1

SLC7A5^ΔCD4
0.070 | 0.37 | 58.3 | 41.3

SLC7A5^WT (-)Leu diet
0.056 | 0.83 | 53.3 | 45.9

7-AAD / AnnV

◀ **Figure EV3.    Impact of dietary L-Leu restriction on T cell activation and survival in aGVHD model.**

(A) Workflow for the aGVHD model with dietary restriction of L-leucine. STC splenic T cell, TCDBM T-cell-depleted bone marrow, Tmt Tomato. (B) Representative images of the macroscopic appearance of $SLC7A5^{WT} \rightarrow$ BALB/c mice fed with regular or (-)Leu diet at 57 d post-transplant. (C) Representative density plots of CD44 vs CD62L gated on allogenic $CD4^+$ and $CD8^+$ cells of spleen from (-)Leu diet-fed $SLC7A5^{WT} \rightarrow$ BALB/c mice and regular-diet fed BALB/c host mice simultaneously injected with $SLC7A5^{\Delta CD4}$ and $SLC7A5^{WT}$ T cells at 7 d post-transplant. (D) Representative density plots of CD69 in allogenic $CD4^+$ (left) and $CD8^+$ (right) cells of spleen from mice described in (C). (E) Representative density plots of AnnV vs 7-AAD gated on total $Tmt^+$ cells of mesenteric lymph nodes from $SLC7A5^{WT} \rightarrow$ BALB/c mice fed with regular or (-)Leu diet and $SLC7A5^{\Delta CD4} \rightarrow$ BALB/c mice at 7 d post-transplant.

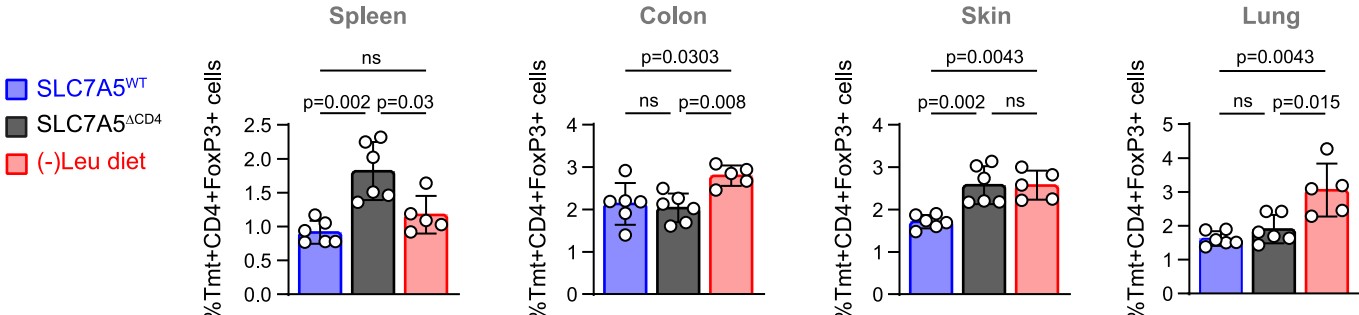

**Figure EV4. Frequency of regulatory T cells.**

Frequencies of FoxP3 cells gated on Tmt⁺CD4⁺ of different tissues from SLC7A5^WT → BALB/c mice fed with regular or (-)Leu diet and SLC7A5^ΔCD4 → BALB/c mice at 7 d post-transplant. $N = 5$–6 per group, mean ± SD, two-tailed Mann–Whitney's test, ns: not significant.

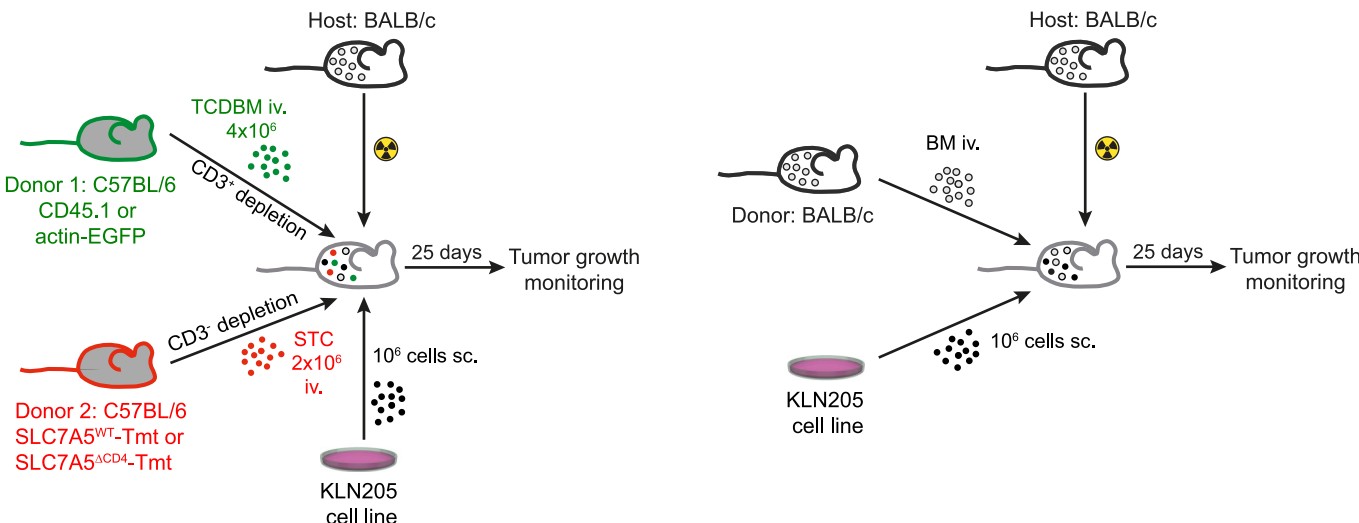

**Figure EV5.  Experimental design to evaluate GVT response.**

Workflow for GVT model comparing the effect of SLC7A5^WT and SLC7A5^ΔCD4 T cells in GVHD groups (left) and control mice (right). STC splenic T cell, TCDBM T-cell-depleted bone marrow, Tmt Tomato.

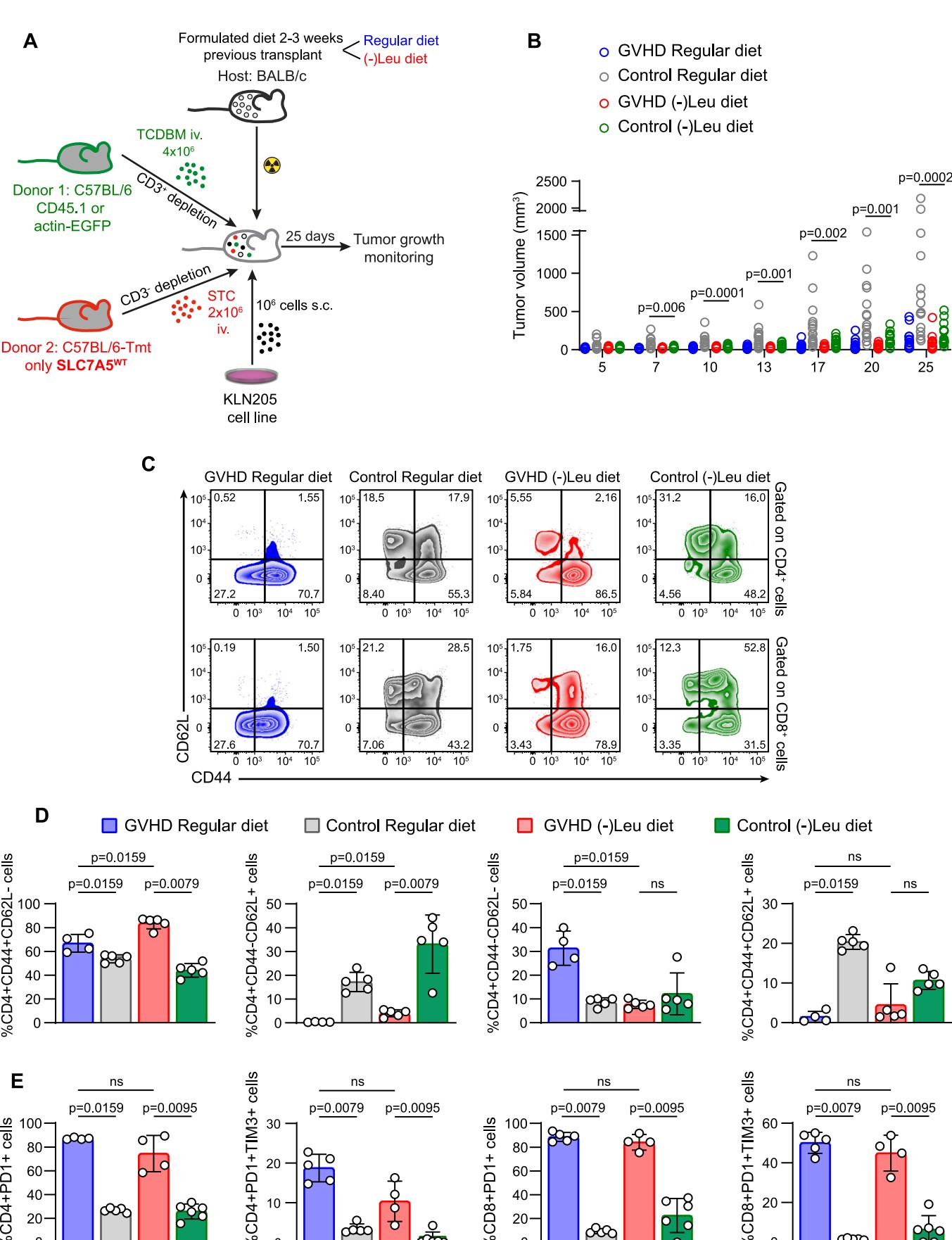

◀ **Figure EV6. Evaluation of dietary L-Leu intake in tumor growth and GVT response.**

(A) Workflow for GVT model comparing the effect of control and (-)Leu diet. STC splenic T cell, TCDBM T-cell-depleted bone marrow, Tmt Tomato. (B) Volume of subcutaneous tumors measured with an electronic calliper. A pool of three independent experiments is shown. Only live animals at the end point are represented: control regular diet $N = 16$, control (-)Leu diet $N = 17$, GVHD regular diet $N = 14$, GVHD (-)Leu $N = 15$. All indicated $p$ values are calculated from control regular diet versus control (-)Leu diet, two-tailed Mann–Whitney's test. (C, D) Representative density plots (C) and frequencies (D) of naive (CD44$^-$CD62L$^+$), effector memory (CD44$^+$CD62L$^-$), double negative (CD44$^-$CD62L$^-$), and central memory (CD44$^+$CD62L$^+$) T cells gated on spleen Tmt$^+$CD4$^+$ and Tmt$^+$CD8$^+$ cells in GVHD groups and CD4$^+$ and CD8$^+$ cells in control groups after 25 d of sc. tumor injection. (E) Frequencies of PD1 and TIM3 gated on Tmt$^+$CD4$^+$CD44$^+$ or Tmt$^+$CD8$^+$CD44$^+$ cells of spleen from GVHD groups and CD4$^+$CD44$^+$ or CD8$^+$CD44$^+$ cells of spleen from control groups after 25 d of sc. tumor injection. (D, E): $N = 4$–5 per group, mean ± SD, two-tailed Mann–Whitney's test.

