## [Peer Review File · EMBO Molecular Medicine]

Restricting SLC7A5-mediated Leucine uptake in T cells prevents acute GVHD and maintains GVT response

Nieves Fernandez-Gallego, Blanca Anega, Susana Luengo-Arias, Maider Bizkarguenaga, Rubén Gil-Redondo, Nieves Embade, Laura Navarrete-Arias, Marta Ramirez-Huesca, Emigdio Álvarez-Corrales, Sara Dosil, Raquel Castillo-González, Amelia Rojas-Gomez, Inés Espeleta, Sara Martínez-Martínez, Arantzazu Alfranca, Virginia G. de Yebenes, Noa Martin-Cófreces, Julián Aragonés, Pilar Martin, Oscar Millet, Francisco Sánchez-Madrid, and Danay Cibrian

Corresponding author: Danay Cibrian (danay.cibrian.externo@salud.madrid.org)

Review Timeline:

Submission Date:	5th Feb 24
Editorial Decision:	29th Feb 24
Appeal:	17th Feb 25
Editorial Decision:	18th Mar 25
Revision Received:	15th Apr 25
Editor's Correspondence:	25th Apr 25
Authors' Correspondence:	29th Apr 25
Accepted:	29th Apr 25

Editor: Lise Roth

Transaction Report:

29th Feb 2024

Decision on your manuscript EMM-2024-19430

Dear Dr. Cibrian,

Thank you for the submission of your manuscript to EMBO Molecular Medicine. We have now received the feedback from the three referees who reviewed your manuscript.

As you will see from the reports below, while they acknowledge the potential interest of the findings, they also raise several major concerns, and are not convinced that the conclusions are supported by the data at this stage.

As clear and conclusive translational insight is key for publication in EMBO Molecular Medicine, and given that at EMBO Press we encourage a single round of revisions in a limited time frame, I am afraid I see little choice but to return the manuscript to you at this point with the decision that we cannot offer to publish it.

I am very sorry to disappoint you in this occasion, and hope that the referees' comments are helpful in your continued work in this area.

With kind regards,

Lise Roth

Lise Roth
Senior Editor
EMBO Molecular Medicine

***** Reviewer's comments *****

Referee #1 (Remarks for Author):

The manuscript by Fernandez-Gallego and colleagues examined the contribution of SLC7A5 and dietary leucine on T cell responses and disease progression in a mouse model of acute graft-versus-host disease (aGVHD). Deletion of SLC7A5, an amino acid transporter of leucine, on T cells mitigated disease progression and rescued mouse survival. Mechanistically, SLC7A5-deficient T cells had impaired proliferation and function, and increased cell death. Restriction of leucine in the diet was sufficient to reduce T cell expansion and promote mouse survival. Overall, the study provides a targeted approach to minimize aGVHD.

Comments:

1. Figure 4A- The authors should include additional markers of activation at early timepoints for the conclusion that TCR-induced activation is not affected in SLC7A5-deficient T cells (see PMID: 23525088).
2. Figure 4C- The reduction in Tbet-expressing CD4+ and CD8+ T cells is relatively modest compared to the pronounced loss in IFN- expression. Is Tbet expression in the cells reduced (i.e., what is the MFI for Tbet)? Are there additional mechanisms that account for the drastic reduction in IFN- expression? Are the expressions of other cytokines affected?
3. Is there an increase in T cell death in mice on leucine-deficient diets, as observed in the absence of SLC7A5, or does impaired proliferation alone account for the reduction in T cells?
4. For the metabolomics analysis, the authors should include non-transplanted, non-irradiated control mice treated with leucine-deficient diets to support that the changes in serum metabolites can be accounted for by the reduction in T cell expansion (as concluded on page 18, lines 413-415) and not by the absence of dietary leucine itself which likely affects more than just the T cells. Further, how is T cell metabolism altered in T cells in mice on leucine-deficient diets? Can the authors get enough cells to perform metabolomics analysis of the T cells, which would support the conclusions made regarding altered TCA cycle and glutaminolysis (page 22, lines 497-499)?

Referee #2 (Comments on Novelty/Model System for Author):

I began reading the manuscript by Nieves Fernández-Gallego with great interest until I reached the results section. The authors state, "Increased neutrophil (CD11b+Ly6C+Ly6G+) and monocyte (CD11b+Ly6C+Ly6G-313) infiltration was detected by flow cytometry in the colon of SLC7A5 WT→BALB/c at 7 days post-transplant, compared to very low infiltration in SLC7A5ΔCD4→BALB/c (Figure 1G)." This suggests less inflammation in the host mice transplanted with T cells from SLC7A5 cKO mice. Although the authors observed fewer donor T cells from SLC7A5 cKO mice, they did not perform competitive migration assays to confirm whether SLC7A5 affects donor T cell migration to GVHD target organs.

In Figure 4, the authors claim that SLC7A5 influences the generation of Tregs. However, they did not propose a mechanism for this process. They also claim that T-bet, which partially controls INF- γ , was decreased, leaving unanswered questions about the cause of this decrease. The authors have not provided data to support that reduced GVHD is due to Tregs, whether these Tregs were functional, or if these cells merely express FoxP3. Missing from these figures is the phenotypic analysis of donor T cells, including activating markers, inhibitory markers, central memory, and, most importantly, effector memory analysis.

In Figure 5, the authors used a CellTrace Violet assay to confirm the proliferation of donor T cells; however, a BrDu or Edu assay would be more appropriate for confirming *in vivo* proliferation on day 7 and day 14. Moreover, the authors performed an Annexin V/7-AAD assay to assess apoptosis, but the data in Fig. 5C shows only ~18% of cells are Annexin positive, with 69% of Annexin V negative cells, which questions the validity of their claim. At this point, my enthusiasm significantly waned.

Upon re-reading the paper to ensure nothing was missed, I question whether the loss of SLC7A5 affects the stemness of these cells. The loss of stemness could explain why the authors did not challenge these mice with tumors or viruses. While I understand that GVT is not always a consequence of bone marrow transplantation, the loss of SLC7A5 appears to shut down the metabolic pathways of these cells. This leads to the question of what the differences are between simply transplanting T and B cell-depleted bone marrow and these cells.

Referee #2 (Remarks for Author):

To improve this paper, the authors could focus on the following areas:

- How SLC7A5 contributes to the metabolism of donor T cells.
- The impact of SLC7A5 on the death of these cells.
- The effect of SLC7A5 on the stemness of these cells, including whether these cells, in naïve, transplant condition, or activation state, can clear tumor or viral infection to substantiate the uncoupling of GVHD from GVT.

Referee #3 (Comments on Novelty/Model System for Author):

The paper is well written and novel because the role of SLC7A5 in GVHD has not been reported. Additional experiments seem to be performed to be qualified as EMBO molecular medicine.

Referee #3 (Remarks for Author):

The authors demonstrate the targeted deletion of SLC7A5 in donor T cells in a GVHD mouse model alleviated the severity of acute GVHD. In addition, reducing L-Leu in diet also regulated immune-mediated injury after an experimental BMT model. The paper is well written and novel because the role of SLC7A5 in GVHD has not been reported. However, the authors did not show whether it could control GVHD as well as maintain GVT effect. So relevant experiments should be performed to be used clinically.

1. In Figure 1G, the authors claimed that neutrophil (CD11b+Ly6C+Ly6G+) and monocyte (CD11b+Ly6C+Ly6G-) infiltration examined by FACS in the colon were increased at 7 days post-transplant in SLC7A5³¹⁴ WT→BALB/c versus in SLC7A5ΔCD4→BALB/c. The recipients of SLC7A5ΔCD4 cells had a very low infiltration of neutrophils and monocytes. However, there is a possibility that the CD11b+Ly6C+Ly6G+ and CD11b+Ly6C+Ly6G- population is MDSC, which protects acute GVHD. How do you explain this? Also, the authors showed that the monocyte population frequency was increased in the SLC7A5ΔCD4 recipients, but the absolute cell count was decreased. Please provide the total cell count.
2. In Figure 2D, The SLC7A5ΔCD4 recipients have more enriched BM-derived or host T cells, as shown in the FACS data. However, the IF data do not indicate a predominance of CD3+ Tmt- cells over CD3+ Tmt+ cells in this group.
3. In Figure 4A, the expression of CD69, a T cell activation marker, was increased in the CD8 T cell population in the SLC7A5ΔCD4 group. Please explain the implications of these results.
4. Figure 5C. CD69 expression on T cells is also different in the CD8 cell populations. Why are only CD4+ results presented? Please add the results of apoptosis in Tmt+CD8+CD69+ cells.
5. Did the authors examine the Treg on donor T cells in the GVHD target organs?
6. Total cell counts of colon, MLN, spleen, skin, and lung are necessary data and should be shown.

As a service to authors, EMBO provides authors with the possibility to transfer a manuscript that one journal cannot offer to publish to another EMBO publication. The full manuscript and if applicable, reviewers reports are automatically sent to the receiving journal to allow for fast handling and a prompt decision on your manuscript. For more details of this service, and to transfer your manuscript to another EMBO title please click on Link Not Available

Point-by-point response

Date: February, 17th, 2025

Manuscript Number: EMM-2024-19430

Former Title of Article: Blockade of SLC7A5 in T-cells by dietary L-Leu restriction prevents acute graft-versus-host disease.

New Article Title: Restricting SLC7A5-mediated Leucine uptake in T cells prevents acute GVHD and maintains GVT response.

Name of the Corresponding Author: Danay Cibrian

Replies to Reviewers' comments

Specific Responses:

Response to Reviewer #1:

***Comment for the authors:** The manuscript by Fernandez-Gallego and colleagues examined the contribution of SLC7A5 and dietary leucine on T-cell responses and disease progression in a mouse model of acute graft-versus-host disease (aGVHD). Deletion of SLC7A5, an amino acid transporter of leucine, on T cells, mitigated disease progression and rescued mouse survival. Mechanistically, SLC7A5-deficient T cells had impaired proliferation and function, and increased cell death. Restriction of leucine in the diet was sufficient to reduce T cell expansion and promote mouse survival. Overall, the study provides a targeted approach to minimize aGVHD.*

We thank the Reviewer for the positive appraisal of our findings.

***Major comment 1:** Figure 4A- The authors should include additional markers of activation at early time points for the conclusion that TCR-induced activation is not affected in SLC7A5-deficient T cells (see PMID: 23525088).*

Following the Reviewer's suggestion, we comparatively analysed the expression of CD62L and CD44 markers in allogenic T cells after seven days of transplant. Our results included in new Figure 3C indicate a reduced expression of CD44 and an increased proportion of naive-fraction in SLC7A5-deficient T cells, suggesting reduced activation. On the other hand, SLC7A5^{WT} T cells do not fail to induce effector memory cells in mice fed without L-Leu (new Figure 3C). We have also analysed the expression of CD62L and CD44 after 25 days in GVT experiments (new Figure 7D, 8G and Supplemental Figure 6C-D), in which similar results were obtained.

***Major comment 2:** Figure 4C- The reduction in Tbet-expressing CD4+ and CD8+ T cells is relatively modest compared to the pronounced loss in IFN- γ expression. Is Tbet expression in the cells reduced (i.e., what is the MFI for Tbet)? Are there additional mechanisms that account for the drastic reduction in IFN- γ expression? Are the expressions of other cytokines affected?*

We have included the mean fluorescence intensity (MFI) data regarding Tbet expression in SLC7A5^{ΔCD4} and SLC7A5^{WT} T cells (new Figure 4A). Tbet upregulation is an earlier event than IFN γ secretion, and SLC7A5 is not expressed in naïve T cells, hence defects in mTORC, glycolysis and protein synthesis that depend on SLC7A5 expression and Leu uptake probably occur after Tbet induction. However, we consider that regulation of Tbet and IFN γ in both experimental settings, SLC7A5 and T cell activation, and the assessment of other cytokines in the absence of L-Leu require further experimentation from the mechanistic and metabolic point of view, but due to space limitations, this should be presented as a new manuscript.

Major comment 3: *Is there an increase in T cell death in mice on leucine-deficient diets, as observed in the absence of SLC7A5, or does impaired proliferation alone account for the reduction in T cells?*

We have performed new experiments to comparatively analyse the Annexin V/7-AAD expression in SLC7A5^{ΔCD4}→BALB/c and SLC7A5^{WT}→BALB/c mice fed with regular diet or (-)Leu diet at 7 days post-transplant. Data shown in the new Figure 3G confirm the reduced cell viability in SLC7A5^{ΔCD4} Tmt⁺ CD4 T cells, and SLC7A5^{WT} cells from mice fed without Leu (Figure 3G and Supplemental Figure 3E).

Major comment 4: *For the metabolomics analysis, the authors should include non-transplanted, non-irradiated control mice treated with leucine-deficient diets to support that the changes in serum metabolites can be accounted for by the reduction in T cell expansion (as concluded on page 18, lines 413-415) and not by the absence of dietary leucine itself which likely affects more than just the T cells. Further, how is T cell metabolism altered in T cells in mice on leucine-deficient diets? Can the authors get enough cells to perform metabolomics analysis of the T cells, which would support the conclusions made regarding altered TCA cycle and glutaminolysis (page 22, lines 497-499)?*

New metabolomics data (New Figure 5) include the following groups: 1) SLC7A5^{WT}→BALB/c mice fed with regular diet; 2) SLC7A5^{WT}→BALB/c mice fed with (-)Leu diet; 3) SLC7A5^{ΔCD4}→BALB/c fed with regular diet; 4) Control (non-irradiated and non-transplanted) BALB/c mice fed with regular diet; and 5) Control (non-irradiated and non-transplanted) BALB/c mice fed with (-)Leu diet. Regarding T cell metabolism, new data included in new Figure 6 show the Seahorse Analysis of cells with genetic deletion of SLC7A5 and cells activated in the absence of L-Leu.

Response to Reviewer #2:

Comment for the authors: I began reading the manuscript by Nieves Fernández-Gallego with great interest until I reached the results section. The authors state, "Increased neutrophil (CD11b+Ly6C+Ly6G+) and monocyte (CD11b+Ly6C+Ly6G-313) infiltration was detected by flow cytometry in the colon of SLC7A5 WT→BALB/c at 7 days post-transplant, compared to very low infiltration in SLC7A5^{ΔCD4} →BALB/c (Figure 1G)." This suggests less inflammation in the host mice transplanted with T cells from SLC7A5^{ΔCD4} mice. Although the authors observed fewer donor T cells from SLC7A5 cKO mice, they did not perform competitive migration assays to confirm whether SLC7A5 affects donor T cell migration to GVHD target organs.

As suggested by the Reviewer, we performed an *in vivo* competitive migration assay (new Supplemental Figure 2B-D and new Figure 2E). This experimental data confirmed the reduced percentage and numbers of CD4⁺ and CD8⁺ SLC7A5^{ΔCD4} cells compared to SLC7A5^{WT} cells in mesenteric lymph nodes, spleen, skin, colon, liver and lung.

In Figure 4, the authors claim that SLC7A5 influences the generation of Tregs. However, they did not propose a mechanism for this process. They also claim that T-bet, which partially controls INF-γ, was decreased, leaving unanswered questions about the cause of this decrease. The authors have not provided data to support that reduced GVHD is due to Tregs, whether these Tregs were functional, or if these cells merely express FoxP3. Missing from these figures is the phenotypic analysis of donor T cells, including activating markers, inhibitory markers, central memory, and, most importantly, effector memory analysis.

We have included additional data concerning naïve/memory markers (new Figure 3C). Analysis of expression of CD62L and CD44 in allogenic T cells indicated a larger proportion of effector memory T cells (CD44⁺CD62L⁻) in the SLC7A5^{WT}→BALB/c group while the SLC7A5^{ΔCD4}→BALB/c mice accumulate more percentage of naïve T cells (CD44⁻CD62L⁺).

Furthermore, we conducted new experimentation to examine FoxP3 expression in the spleen and further GVHD target tissues, in SLC7A5^{ΔCD4}→BALB/c and SLC7A5^{WT}→BALB/c mice fed with or without Leu diet, at 7 days post-transplant (new Supplemental Figure 4). The increase of CD4⁺Foxp3⁺ T cells percentage in the absence of SLC7A5 or (-)Leu diet was detected in some but not in all tissues. We agree with the Reviewer that we have not analysed whether these Tregs were functional or if these cells merely express FoxP3, both in SLC7A5^{ΔCD4} and (-)Leu deprivation. In addition, we did not remove FoxP3⁺ cells in the purification of the STC from donor mice, thus the possibility that natural Treg cells could be detected after 7 days of transplant cannot be ruled out. We consider that these aspects require extensive additional experimentation; therefore, it may be the focus of a future work. For this reason, data regarding CD4⁺FoxP3⁺ T cell detection is shown as supplemental information in the revised manuscript, and the limitations of our research and conclusion are commented in the results section.

In Figure 5, the authors used a CellTrace Violet assay to confirm the proliferation of donor T cells; however, a BrDu or Edu assay would be more appropriate for confirming *in vivo* proliferation on day 7 and day 14. Moreover, the authors performed an AnnV/7-AAD assay to assess apoptosis, but the data in Fig. 5C shows only ~18% of cells are Annexin positive, with 69% of AnnV negative cells, which questions the validity of their claim. At this point, my enthusiasm significantly waned.

Cell proliferation has been determined *in vivo* with BrdU at 7 days post-transplant in SLC7A5^{WT}→BALB/c and SLC7A5^{ΔCD4}→BALB/c mice. Results shown below (Figure A for Reviewers) confirmed a significant reduction of T cell proliferation in SLC7A5^{ΔCD4} T cells, at 7d post-transplant. Due to space limitations, we do not include these data in the revised manuscript.

Figure A. Percentages (left) and absolute numbers (right) of BrdU in Tmt⁺CD4⁺ (top) and Tmt⁺CD8⁺ from spleen of SLC7A5^{WT}→BALB/c and SLC7A5^{ΔCD4}→BALB/c mice at 7 d post-transplant. N=5 per group, mean±SD, two-tailed Mann-Whitney's test, *p≤0.05, **p≤0.01.

Upon re-reading the paper to ensure nothing was missed, I question whether the loss of SLC7A5 affects the stemness of these cells. The loss of stemness could explain why the authors did not challenge these mice with tumors or viruses. While I understand that GVT is not always a consequence of bone marrow transplantation, the loss of SLC7A5 appears to shut down the metabolic pathways of these cells. This leads to the question of what the differences are between simply transplanting T and B cell-depleted bone marrow and these cells.

Following the Reviewer's suggestion, GVT experiments have been performed and included as new Figure 7 (new Supplemental Figure 5) and new Figure 8 (new Supplemental Figure 6). The absence of SLC7A5 expression indeed compromised GVT response but, surprisingly, our data demonstrate that Leu-deprivation from diet controls GVHD and further supports GVT response.

Major comment 1: How SLC7A5 contributes to the metabolism of donor T cells.

We agree with the Reviewer that unravelling how SLC7A5 and L-Leu deprivation controls T cell metabolism is an interesting issue to be investigated. We have addressed this *in vitro* using Seahorse Real-Time Cell Metabolic Analysis (new Figure 6). Our data show that both energy-producing pathways, glycolysis and OXPHOS, are severely impaired in SLC7A5^{ΔCD4} cells. However, L-Leu deficiency selectively reduces glycolysis but maintains the OXPHOS function of activated T cells. Hence, our manuscript highlights the effect of Leu uptake through SLC7A5 in the role of glycolysis and underscores Leu-independent mechanisms by which OXPHOS is affected in SLC7A5-deficient T cells.

Major comment 2: The impact of SLC7A5 on the death of these cells.

We have performed new experiments to assess the role of SLC7A5-deficiency and L-Leu-withdrawal in cell death (new Figure 3G and Supplemental Figure 3E). Interestingly, our results underscore a differential sensitivity to cells death between CD4 and CD8 T cells. The genetic deletion of SLC7A5 as well as L-Leu deprivation increase CD4 T cells apoptosis *in vivo* (new Figure 3G and Supplemental Figure 3E). On the contrary, SLC7A5^{ΔCD4} CD8⁺ T cells were protected from apoptosis and no differences were detected in SLC7A5^{WT} cells in mice fed with or without L-Leu. The results of reduced CD8⁺ T cell apoptosis is probably due to reduced expression of granzyme B (PMID: 16547231). We have also performed *in vitro* experiments where similar results were observed, but due to space limitation, they will not be included in the revised manuscript (see Figure C for reviewer 3).

Major comment 3: The effect of SLC7A5 on the stemness of these cells, including whether these cells, in naïve, transplant condition, or activation state, can clear tumor or viral infection to substantiate the uncoupling of GVHD from GVT.

New experiments were conducted to study the effect of SLC7A5 genetic deletion in T cells (new Figure 7 and new Supplemental Figure 5) and dietary restriction of L-Leu (new Figure 8 and new Supplemental Figure 6) in *in vivo* models of GVT response. These data demonstrate that SLC7A5-deficient T cells were not able to clear the tumor while dietary restriction of L-Leu contributed to reducing tumor growth and retaining of GVT response by allogenic T cells. Our data indicate that CD8⁺ T cells can be activated in the absence of Leu, maintaining IFN γ expression and increasing the proportion of central memory T cells, which could be supporting GVT response. In addition, we observed that the administration of an L-Leu-deprived diet increases the expression of the stemness marker TCF1 in T cells. We consider that our data represent a novel and relevant contribution, in addition to raising many questions regarding the relevance of L-Leu uptake or L-Leu-derived metabolites to sustain the stemness of T cells and GVT response.

Response to Reviewer #3:

Comment for the authors: The paper is well written and novel because the role of SLC7A5 in GVHD has not been reported. Additional experiments seem to be performed to be qualified as EMBO molecular medicine.

The authors demonstrate the targeted deletion of SLC7A5 in donor T cells in a GVHD mouse model alleviated the severity of acute GVHD. In addition, reducing L-Leu in diet also regulated immune-mediated injury after an experimental BMT model. The paper is well written and novel because the role of SLC7A5 in GVHD has not been reported. However, the authors did not show whether it could control GVHD as well as maintain GVT effect. So relevant experiments should be performed to be used clinically.

We thank the Reviewer for the kind words and positive comments. As requested, new data regarding the role of SLC7A5-genetic deletion in allogenic T cells (new Figure 7 and Supplemental Figure 5) and (-)Leu diet (new Figure 8 and Supplemental Figure 6) in GVT response has been included in the manuscript.

Major comment 1: In Figure 1G, the authors claimed that neutrophil (CD11b+Ly6C+Ly6G+) and monocyte (CD11b+Ly6C+Ly6G-) infiltration examined by FACS in the colon were increased at 7 days post-transplant in SLC7A5^{WT}→BALB/c versus in SLC7A5^{ΔCD4}→BALB/c. The recipients of SLC7A5^{ΔCD4} cells had a very low infiltration of neutrophils and monocytes. However, there is a possibility that the CD11b+Ly6C+Ly6G+ and CD11b+Ly6C+Ly6G- population is MDSC, which protects acute GVHD. How do you explain this?

Also, the authors showed that the monocyte population frequency was increased in the SLC7A5^{ΔCD4} recipients, but the absolute cell count was decreased. Please provide the total cell count.

We appreciate the Reviewer's comment. The increase in CD11b⁺ cells detected in the colon correlated with increased damage of the epithelial layer detected in mice transplanted with SLC7A5^{WT} cells, after 7 days. We agree with the Reviewer that the study of MDSCs will be interesting in SLC7A5^{ΔCD4}→BALB/c mice or (-)Leu deficient diet, due to their immunosuppressive properties and their relevance in the GVHD and GVT response. However, we consider that exploring this possibility is beyond the scope of the article as we have focused it on the effect of SLC7A5-mediated Leu uptake by T cells.

Regarding frequency and cell count differences, it is explained because cell counting is the absolute number of cells detected in the full-length colon, whereas the frequency is a relative value that depends on the gating strategy. In the case of the former Figure 1G, now included as new Supplemental Figure 1C, data is from gated colon live cells CD45.2⁺CD11b⁺ myeloid population. Hence, in SLC7A5→BALB/c mice where tissue injury is higher, the increased infiltration of neutrophils simultaneously decreases the percentage of Ly6C⁺Ly6G⁻ cells (including macrophages and monocytes) from the total of gated CD11b⁺ cells. The total cell

count of CD45.2⁺CD11b⁺ cells is shown in Figure B for the reviewer, but due to space restriction, we have decided not to include it in the revised form of the manuscript.

Figure B. Total number of myeloid cells (CD45.2⁺CD11b⁺) in colon of SLC7A5^{WT}→BALB/c and SLC7A5^{ΔCD4}→BALB/c mice at 7 d post-transplant. N=6 per group, mean±SD, two-tailed Mann-Whitney's test, *p≤0.05, **p≤0.01.

Major comment 2: In Figure 2D, the SLC7A5^{ΔCD4} recipients have more enriched BM-derived or host T cells, as shown in the FACS data. However, the IF data do not indicate a predominance of CD3⁺ Tmt⁻ cells over CD3⁺ Tmt⁺ cells in this group.

New Figure 1I includes double staining of Tmt⁺ (allogenic CD3) and GFP⁺ (BM-derived) cells in the spleen of SLC7A5^{ΔCD4}→BALB/c and SLC7A5^{WT}→BALB/c mice.

Major comment 3: In Figure 4A, the expression of CD69, a T cell activation marker, was increased in the CD8 T cell population in the SLC7A5^{ΔCD4} group. Please explain the implications of these results.

We have performed new experimentation to comparatively analyse the expression of CD69 in SLC7A5^{ΔCD4}→BALB/c and SLC7A5^{WT}→BALB/c mice fed with or without Leu diet, after 7 d of transplant (new Figure 3D). Our results confirm the increase of CD69 percentage in SLC7A5^{ΔCD4} T cells, both CD4⁺ and CD8⁺ T cells. These results agree with previous data reported by Sinclair *et al* (PMID: 23525088) in which increased CD69 expression was observed in activated CD8⁺ T cells. Our new data also show that deprivation of Leu also increases the percentage of CD69 expression. A possible explanation for this effect was included in the discussion section.

Major comment 4: Figure 5C. CD69 expression on T cells is also different in the CD8 cell populations. Why are only CD4⁺ results presented? Please add the results of apoptosis in Tmt⁺CD8⁺CD69⁺ cells.

We analyse the Annexin V/7-AAD expression in SLC7A5^{ΔCD4}→BALB/c and SLC7A5^{WT}→BALB/c mice fed with regular or (-)Leu diet at 7 days post-transplant. Data shown as new Figure 3G confirm the reduced cell viability in SLC7A5^{ΔCD4} Tmt⁺ T cells, and SLC7A5^{WT} cells from mice fed

without Leu. Importantly, the effect of increased cell apoptosis by deletion of SLC7A5 or L-Leu withdrawal from the diet is mainly detected in CD4⁺ T cells, whereas SLC7A5^{ΔCD4} CD8⁺ T cells were protected from apoptosis and no differences were detected in SLC7A5^{WT} cells in mice fed without L-Leu.

The results obtained with *in vitro* activated CD4 and CD8 T cells from SLC7A5^{ΔCD4} and SLC7A5^{WT} were similar to *in vivo* results (see below Figure C for the Reviewer). However, we consider that the results of cell apoptosis obtained *in vivo* in the GVHD model are more significant for the manuscript than results observed with *in vitro* antiCD3/CD28-activated T cells, and for that reason, we have decided to exclude *in vitro* data from the revised manuscript.

Figure C. CD4 and CD8 cells were purified from spleen and lymph nodes of SLC7A5^{WT} and SLC7A5^{ΔCD4} mice and activated with anti-CD3/anti-CD28 for 48 h. Representative density plots (left) and frequencies (right) of AnnV⁺ cells gated on Tmt⁺CD69⁺CD4⁺ and CD8⁺ cells. N=5 per group, mean±SD, two-tailed Mann-Whitney's test, *p<0.05, **p<0.01.

Major comment 5: Did the authors examine the Treg on donor T cells in the GVHD target organs?

We comparatively analyse CD4⁺FoxP3⁺ T cells in GVHD target tissues at 7 days post-transplant in SLC7A5^{ΔCD4}→BALB/c and SLC7A5^{WT}→BALB/c mice fed with or without Leu diet. New data is included as new Supplemental Figure 4. The increase of CD4⁺Foxp3⁺ T cells percentage in the absence of SLC7A5 or with (-)Leu diet was detected in some but not in all tissues. However, as

we did not remove FoxP3⁺ cells from STC of donor mice and we have not performed a functional analysis of the regulatory capacity of SLC7A5^{ΔCD4} CD4⁺Foxp3⁺ or SLC7A5^{WT} CD4⁺Foxp3⁺ detected in (-)Leu deficient diet, we cannot conclude that SLC7A5 or (-)Leu deficiency influence Treg function. We consider this to be beyond the scope of the current manuscript.

Major comment 6: *Total cell counts of colon, MLN, spleen, skin, and lung are necessary data and should be shown.*

As indicated by the Reviewer, we have included total cell numbers of the spleen and mesenteric lymph nodes as new Supplemental Figure 1E, and of the remaining organs as new Supplemental Figure 2A.

18th Mar 2025

Dear Dr. Cibrian,

Thank you for submitting your revised study. We had initially rejected your manuscript post-review, as despite the potential interest of the results noted by the referees, the requested revisions did not seem realistic in a 3-6 months time frame. You have nevertheless chosen to revise the manuscript according to the referees' concerns, and after consideration of the new results and discussion within the team, we sent it back to the referees. We have now received the reports from referees #1 and #3, who also evaluated your responses to referee #2. As you will see below, they are satisfied with the revisions, and I will therefore be able to accept your manuscript once the following editorial comments are addressed:

1/ Manuscript text:

- Please remove the blue font text and only keep in track changes mode any new modification.
- The list of abbreviations should be removed from the manuscript text.
- Please remove "data not shown" (p. 19). As per journal policy, all discussed results must be shown in the main or supplementary figures.
- "Materials and Methods" should be renamed "Methods" and placed after the discussion:
 - o Please provide a Reagents and Tools Table (listing key reagents, experimental models, software and relevant equipment and including their sources and relevant identifiers). Kindly download and fill our Reagents and Tools Table template (.docx), which you can find in our author guidelines: <https://www.embopress.org/page/journal/14693178/authorguide#structuredmethods>. When submitting your revised manuscript, do not include the Reagents and Tools Table in the Methods section of the manuscript but upload it as a separate file choosing the file type "Reagent Table".
 - o Cells: please provide a statement on mycoplasma contamination.
 - o Antibodies: please provide dilutions/concentrations.
 - o Statistics: please provide a statement on randomization, blinding, sample size and inclusion/exclusion criteria.
- Data availability: please remove "All data and reagents are available from the corresponding author upon request." Please only list here accession numbers and databases for primary datasets produced in this study and deposited in an appropriate public database. In case you have no data that requires deposition in a public database, please state so in this section (This study includes no data deposited in external repositories). This section should be placed after the Methods section.
- Author contributions: CRediT has replaced the traditional author contributions section because it offers a systematic machine readable author contributions format that allows for more effective research assessment. Please remove the Authors Contributions from the manuscript and use the free text boxes beneath each contributing author's name in our system to add specific details on the author's contribution. More information is available in our guide to authors.
- "Conflict of interest" should be renamed "Disclosure statement and competing interests": Please review the policy <https://www.embopress.org/competing-interests> and update your competing interests if necessary. This section should be placed after "Acknowledgements".
- Funding and Acknowledgements: the "funding" section on the first page of the manuscript should be removed. The "financial support" section should be merged with Acknowledgments and the list should match the list that is entered into the submission system.
- References: please correct the format from numerical to alphabetical order, and list 10 author names before et al.

2/ Figures:

- Exact p values should be provided in the figures or their legends.
- The supplementary figures should be renamed "Figure EV1-6". Their legends should be after the main figure legends, under the heading "Expanded View Figure Legends"
- Please note that there is a callout for a Table 1 in the text, but no such table has been provided.
- Table S1 should be renamed "Table EV1", removed from the manuscript text and uploaded as a separate file
- Please address the queries from our copy editors in the figure legends:
 1. Please note that the legends for figure 2 is not provided in the sequential manner (legend for figure 2C, D is provided before legend of figure 2B). This needs to be rectified.
 2. Please note that the exact p values are not provided in the legends of figures 1A, B, C, F, G, H; 2A, C, D, E; 3A-G; 4A-E; 5B, 6A, B; 7A-E; 8A, B, C, F, G, H; supplementary figures S1 E, S2 A, C, D; S4, S6 B, E
 3. Please note that information related to n is missing in the legend of figure 5B
 4. Please note that the error bars are not defined in the legend of figure 5B.

3/ At EMBO Press we ask authors to provide source data for the main figures. Our source data coordinator will contact you to discuss which figure panels we would need source data for and will also provide you with helpful tips on how to upload and organize the files.

4/ Please provide a complete author checklist, which you can download from our author guidelines (<https://www.embopress.org/page/journal/17574684/authorguide#submissionofrevisions>). Please insert information in the

checklist that is also reflected in the manuscript. The completed author checklist will also be part of the RPF.

5/ Please provide "The paper explained": EMBO Molecular Medicine articles are accompanied by a summary of the articles to emphasize the major findings in the paper and their medical implications for the non-specialist reader. Please provide a draft summary of your article highlighting

6/ Please provide a "Synopsis" text: Synopses are displayed on the journal webpage and are freely accessible to all readers. They include a short stand first (maximum of 300 characters, including space) as well as 2-5 one-sentences bullet points that summarizes the paper. Please write the bullet points to summarize the key NEW findings. They should be designed to be complementary to the abstract - i.e. not repeat the same text. We encourage inclusion of key acronyms and quantitative information (maximum of 30 words / bullet point). Please use the passive voice. Please attach these in a separate file or send them by email, we will incorporate them accordingly.

Please also suggest a visual abstract to illustrate your article as a PNG file 550 px wide x 300-600 px high. A cropped portion of this image will serve as thumbnail for the table of content on our webpage.

7/ As part of the EMBO Publications transparent editorial process initiative (see our Editorial at <http://embomolmed.embopress.org/content/2/9/329>), EMBO Molecular Medicine will publish online a Review Process File (RPF) to accompany accepted manuscripts.

This file will be published in conjunction with your paper and will include the anonymous referee reports, your point-by-point response and all pertinent correspondence relating to the manuscript. Let us know whether you agree with the publication of the RPF and as here, if you want to remove or not any figures from it prior to publication.

I look forward to receiving your revised manuscript.

Yours sincerely,

Lise Roth

***** Reviewer's comments *****

Referee #1 (Remarks for Author):

The authors have largely addressed my comments, and those comments from reviewer 2.

Referee #3 (Comments on Novelty/Model System for Author):

This manuscript reveals a novel mechanism of graft-versus-host disease following experimental allogeneic stem cell transplantation. The experimental design is scientifically relevant and well-detailed. The authors have responded thoroughly to the reviewers' queries.

Referee #3 (Remarks for Author):

The authors have responded thoroughly to the reviewers' queries.

The authors addressed the minor editorial issues.

Dear Dr. Cibrian,

Thank you for submitting your revised files. I have gone through all the folders and am now ready to accept your manuscript.

However, before I can do so, please check Figure 7E and the associated raw data carefully (the two panels on the right are identical). Please correct if necessary (including source data) and provide an explanation.

I have cropped a small part of your synopsis image to serve as a thumbnail on our website (attached), please let me know if you agree or provide another image (115x70 pixels).

Finally, we understand from your correspondence that you would like all figures from the point-by-point rebuttal to be removed from the review process file, could you please confirm that this is correct?

We look forward to hearing from you,

With kind regards,

Lise Roth

Dear Dr. Roth,

We thank the editor for pointing out the error in Figure 7E. Indeed the data corresponding to total PD1 expression in CD8 T cells is correct, but the graph corresponding to CD4 is a duplicate of CD8. This was an unfortunate mistake when assembling the figure, we duplicated the layout from CD8 in the graphpad to maintain the format, but we did not replace the data corresponding to CD4.

We are sending you Figure 7 with the correct panels and the source data for CD4 and CD8 populations in Figure 7E included in the new zip folder for Figure 7. We apologize once again for the mistake during the preparation of the figures. Please let me know if we have to update the corrected Figure 7 and source data_Figure 7 on the submission website. The result and p-value do not change.

We would like to propose an alternative image for the thumbnail.

We agree with the publication of all figures in the point by point. We're sorry if we haven't been clear in our previous communication.

We would appreciate it if you could let us know when you've received the files. We are still having issues with the internet connection due to the power failure.

Best,

Danay

29th Apr 2025

Dear Dr. Cibrian,

Thank you for submitting your revised files and clarifying the issue with Figure 7. I am pleased to inform you that your manuscript is accepted for publication and is now being sent to our publisher to be included in the next available issue of EMBO Molecular Medicine.

With kind regards,

Lise Roth
